# FROM EVIDENCE TO TRAJECTORY: ABDUCTIVE REASONING PATH SYNTHESIS FOR TRAINING RETRIEVAL-AUGMENTED GENERATION AGENTS

## ABSTRACT

Retrieval-augmented generation agents development is hindered by the lack of process-level supervision to effectively guide agentic capabilities like task decomposition, retriever invocation, and stepwise decision-making. While reinforcement learning offers a potential solution, it suffers from sparse rewards and the limited reasoning capabilities of large language models (LLMs). Meanwhile, existing data synthesis methods only produce chain-of-thought rationales and fail to model environmental interactions. In this paper, we propose **EviPath**, an evidence-anchored reasoning path synthesis paradigm for RAG agent development. EviPath comprises: (i) Abductive Subtask Planning, which decomposes the problem into sub-questions and iteratively plans an optimal solution path based on the dependencies between them; (ii) Faithful Sub-question Answering, which uses supporting evidence to construct a proxy environment to generate reasoning thoughts and answers for each sub-question; and (iii) Conversational Fine-Tuning, which formats the complete agent-environment interaction trajectory into a dialogue format suitable for Supervised Fine-Tuning. EviPath allows LLMs to learn complex reasoning and tool-use capabilities directly from synthesized data. Extensive experiments on widely-used question-answering benchmarks show that an 8B parameter model trained with EviPath-synthesized data significantly and consistently outperforms state-of-the-art baselines with **a double-digit absolute EM gain of 14.7%** in open-domain question answering.

## 1 INTRODUCTION

Retrieval-augmented generation (RAG) agents, powered by large language models (LLMs) (Guo et al., 2025), can autonomously gather external knowledge and answer complex, multi-hop questions. Compared to vanilla RAG systems (Lewis et al., 2020), RAG agents minimize the need for human intervention, and adapt readily to downstream applications like math problem solving (Zhu et al., 2025), code generation (Zhang et al., 2023), and financial analysis (Wang et al., 2025c).

Despite their promise, RAG agents are hard to develop since ***ground truth reasoning trajectories are unavailable***. Mainstream multi-hop question answering datasets Yang et al. (2018); Ho et al. (2020); Trivedi et al. (2022) provide final answers and supporting facts, while lacking step-wise supervision that is crucial to equip LLMs with agentic behaviors like question decomposition, search query reformulation, and plan refinement. As a result, existing RAG agents (Li et al., 2025; Xu et al., 2025) still fail to deliver reliable performance.

One approach to train RAG agents without process supervision is reinforcement learning (RL) (Shao et al., 2024), which optimizes the decision-making process based on outcome-based rewards. Nevertheless, these methods have notable limitations. First, reward signals are often sparse and delayed, making it difficult to assign credit to individual decisions. In addition, RAG agents contain various non-differentiable components like retrievers and databases, which makes end-to-end gradient backpropagation infeasible. Most critically, the effectiveness of RL relies heavily on the intrinsic reasoning capabilities of the model. ***Without sufficient prior knowledge, the LLM may fail to discover correct actions that lead to a positive reward, rendering trajectory exploration ineffective***.

Figure 1: The limitations of reinforcement learning- or data synthesis-based approaches in training RAG agents to answer open-domain questions.

Another line of work (Bai et al., 2024; Yang et al., 2025) mitigates data scarcity by synthesizing reasoning paths, which leverage LLMs to generate chain-of-thought (CoT) rationales that link questions to answers using the supporting evidence. However, existing data synthesis approaches cannot be adapted to agent training. First, the generated paths often function as post-hoc explanations of predefined answers, rather than a genuine, step-by-step problem-solving process. More importantly, *simple CoT training does not endow RAG agents with the core agentic capability to interact with external environments*, severely limiting their effectiveness in answering open-domain questions.

Considering the aforementioned limitations, we propose **EviPath**, an **Evi**dence-anchored reasoning **path** synthesis framework based on *abductive reasoning*. **EviPath** proceeds in three stages that align with the Planner-Executor architecture (Li et al., 2025) of RAG agents: Firstly, the *(i) Abductive Subtask Planning* stage applies abductive reasoning on the final answer and supporting evidence to reverse-engineer an optimal, dependency-aware reasoning plan, and then simulates the agent's iterative execution process to generate thoughts and retrieval queries that form the planner's reasoning path. Secondly, the *(ii) Faithful Sub-question Answering* stage operates in a simulated environment to bypass retrieval errors, where it identifies the exact evidence for each sub-question to synthesize grounded thoughts and derive an intermediate answer. Finally, the *(iii) Conversational Fine-tuning* stage packages the complete reasoning paths from previous steps into a user-assistant dialogue format for supervised fine-tuning (SFT). Extensive experiments on widely used *open-domain* QA benchmarks show that an 8B-RAG agent fine-tuned with EviPath trajectories significantly outperforms state-of-the-art baselines, achieving **a double-digit absolute EM gain of 14.7%**. Our contributions can be summarized in the following four aspects:

- We are the first to formulate the synthesis of reasoning paths for RAG agents as an *abductive reasoning problem*. This novel perspective provides a structured approach for generating interactive, goal-oriented reasoning trajectories.

- Building on this formulation, we propose **EviPath**: a novel framework that synthesizes reasoning paths to solve the dual challenges of data scarcity and reliance on complex reinforcement learning, *establishing a data-centric paradigm for RAG agent development*.

- We construct **265k** golden reasoning paths from multi-hop QA benchmarks, specifically designed to enhance agentic skills like high-level planning, retriever use, and context-aware reasoning.

- We conduct extensive experiments on three widely used multi-hop QA datasets. The results show that RAG agents trained on EviPath-synthesized data significantly and consistently outperform all state-of-the-art retrieval-augmented generation agents *in open-domain settings*.

## 2 RELATED WORKS

### 2.1 RAG AGENTS FOR QA

RAG agents enhance LLMs with external evidence to mitigate hallucinations in knowledge-intensive QA (Achiam et al., 2023; Cheng et al., 2024; Huang et al., 2025). The paradigm has evolved from simple "retrieve-then-read" pipelines (Lewis et al., 2020; Gao et al., 2023) to sophisticated work-

flows that interleave reasoning, tool use, and reflection (Yao et al., 2023; Trivedi et al., 2023; Asai et al., 2024; Shao et al., 2023). Recent works scale agentic QA through learned monologues (Yang et al., 2024), and modular designs that separate planning from execution (Li et al., 2025; Xu et al., 2025; Jiang et al., 2025a). Some efforts also leverage reinforcement learning to optimize policies of retrieval and reasoning (Jin et al., 2025; Song et al., 2025a; Wu et al., 2025). Orthogonal advances strengthen the retrieval side, including query reformulation (Chan et al., 2024; Mao et al., 2024), end-to-end multi-hop retrieval (Zhang et al., 2024a), and knowledge graph integration (Luo et al., 2025; Wang et al., 2025a; Hao et al., 2025). Despite making great progress, state-of-the-art methods seldom expose fine-grained process supervision that our work introduces to precisely guide LLMs in performing agentic operations like question decomposition, retrieval invocations, and reasoning.

## 2.2 Reasoning Path Synthesis

Data synthesis is common for enhancing the reasoning capabilities of LLMs (Wang et al., 2023; Xiong et al., 2024; Bai et al., 2024). Early methods leverage LLMs for direct generation of reasoning paths (Yu et al., 2023; Wang et al., 2023), but these were often ungrounded and prone to hallucination. To improve faithfulness, subsequent work has synthesized training data for long-context reasoning by incorporating citations to ground truth evidence (Yang et al., 2025; Bai et al., 2024) or by concatenating long training sequences from existing contexts (Xiong et al., 2024; Gao et al., 2025; An et al., 2024). However, these approaches primarily generate reasoning chains over static, predefined contexts, failing to guide the training of RAG agents that necessitate extensive environment interactions. For detailed analysis of related works, please refer to Appendix A.

## 3 Problem Formulation

### 3.1 Answering Questions with RAG Agents

In this paper, we develop RAG agents to address the multi-hop question answering (MHQA) task in an open-domain setting. The core challenge of MHQA lies in aggregating evidence from diverse sources and conducting multi-step reasoning to derive the final answer. To address this, RAG agents (Figure 2) decouple the complex reasoning process into a hierarchical, two-level framework, which consists of a **Planner** for high-level strategic planning and an **Executor** for low-level sub-task execution.

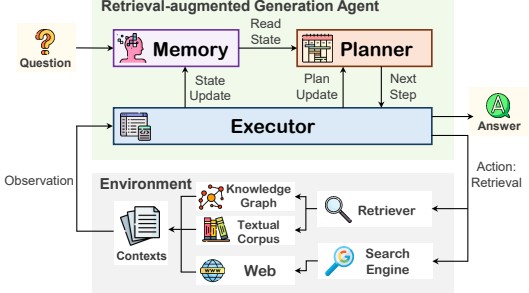

Figure 2: The architecture of RAG agents.

The overall process is as follows: given a complex question $q$, the high-level **Planner** first formulates a plan $\mathcal{P}$ by decomposing $q$ into a sequence of atomic, solvable sub-questions. Subsequently, the plan is executed in an iterative fashion. At each reasoning step $i$, the **Planner** determines the specific set of sub-questions $Q_i$ to be resolved in the current step based on the progress made so far. Then, the low-level **Executor** takes charge of each sub-question $q_j \in Q_i$, interacts with an external knowledge base $\mathcal{K}$ to retrieve relevant context $\mathcal{C}_j$, and derives an answer $a_j$ for that sub-question. This process continues until all sub-questions are addressed. Finally, the **Planner** synthesizes the intermediate results into a final answer $a$. Formally, the collaborative workflow can be expressed as:

$$P(a|q,\mathcal{I},\mathcal{K}) = \underbrace{P(\mathcal{P}|q,\mathcal{I}) \cdot \prod_{i=1}^{|\mathcal{P}|} P\big(Q_i,\mathcal{R}_i \mid Q_{<i}, a_{<i}, q, \mathcal{P}\big)}_{\text{Planner}} \cdot \underbrace{\big(\prod_{q_j \in Q_i} P(\mathcal{C}_j|q_j,\mathcal{K}) \cdot P(a_j,\mathcal{R}_j^{(s)}|q_j,\mathcal{C}_j)\big)}_{\text{Executor}}, \quad (1)$$

where $\mathcal{I}$ denotes the instruction prompts, $|\mathcal{P}|$ denotes the number of reasoning steps, $a_{<i}$ contains the answer of all sub-questions prior to the $i$-th reasoning step; $\mathcal{R}_i$ denotes the reasoning thoughts made in $i$-th planning step, $\mathcal{R}_j^{(s)}$ denotes the thoughts for answering sub-question $q_j$.

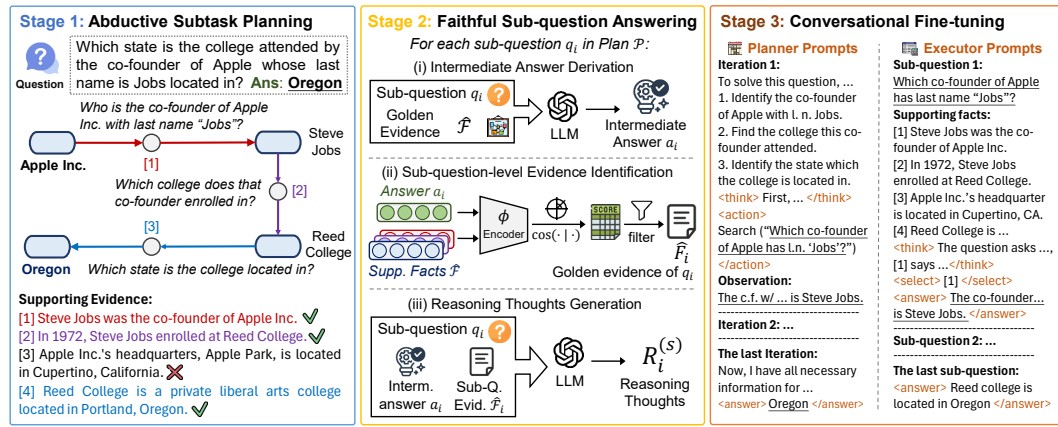

Figure 3: The end-to-end workflow of the proposed EviPath framework.

## 3.2 REASONING PATH SYNTHESIS AS AN ABDUCTIVE REASONING TASK

We consider an MHQA dataset to consist of training problems in the form of $d_{\text{train}} = (q, a, \mathcal{F}, \hat{\mathcal{F}})$, where $q$ is the question, $a$ is the answer, $\mathcal{F}$ represents the pool of supporting facts with distractors, and $\hat{\mathcal{F}}$, the set of golden evidence. We argue that **reasoning path synthesis** constitutes an **abductive reasoning** (Josephson & Josephson, 1996) task, which aims to infer a trajectory $\mathcal{T}_{q \to a}$ that best derives how the observed outcome (answer) $a$ follows from the premise (question) $q$ under the constraints and evidential support provided by $\hat{\mathcal{F}} \subsetneq \mathcal{F}$. Formally, we have the objective function:

$$\mathcal{T}_{q \to a} = (\mathcal{P}, \{\mathcal{R}_i\}_{i=1}^{|\mathcal{P}|}, \{Q_i\}_{i=1}^{|\mathcal{P}|}, \{A_i\}_{i=1}^{|\mathcal{P}|}, \{\mathcal{R}_j^{(s)}\}_{j=1}^N) = f_{\text{LLM}}(q \wedge a | \hat{\mathcal{F}}, \mathcal{F}), \qquad (2)$$

where $A_i = \{a_j | q_j \in Q_i\}$ is the set of answers to the sub-questions in step $i$, $N = \sum_i |Q_i|$ is the total number of sub-questions to be resolved, $f_{\text{LLM}}$ denotes the LLM. Since $\hat{\mathcal{F}}$ provides ground truth evidence that is minimally sufficient to answer the question $q$, conducting abductive reasoning process based on $\hat{\mathcal{F}}$ yields concise reasoning trajectories with fewer reasoning steps, while remaining alignment to the expected outcome $a$.

## 4 METHOD

In this section, we present EviPath (Figure 3), a reasoning-path synthesis framework aligned with the planner–executor architecture of RAG agents. The pipeline comprises two phases: *(i) Abductive Subtask Planning (ASP)* and *(ii) Faithful Sub-question Answering (ESA)*, each of which corresponds to the Planner and Executor modules, respectively. We then leverage these complete reasoning path to develop and refine RAG agents via *(iii) conversational fine-tuning (CFT)*.

### 4.1 ABDUCTIVE SUBTASK PLANNING (PLANNER-SIDE REASONING PATH SYNTHESIS)

#### 4.1.1 TASK DECOMPOSITION

To solve a complex question $q$, the RAG agent first decomposes it into a plan $\mathcal{P}$ with a set of sub-questions $\{q_1, q_2, \cdots, q_n\}$. The quality of this initial plan is crucial, as it constrains the search space and improves overall accuracy and efficiency of reasoning. However, despite having strong semantic understanding capabilities, LLMs often fail to generate coherent multi-step plans without direct supervision. This challenge is compounded by the fact that mainstream QA datasets offer the final answer $a$, supporting facts $\mathcal{F}$, and a golden evidence subset $\hat{\mathcal{F}}$, but lack golden question decomposition for LLM fine-tuning. To bridge the aforementioned supervision gap, EviPath introduces **abductive reasoning** to reverse-engineer a latent reasoning graph by analyzing the ground-truth answer and the dependencies between different pieces of evidence with an LLM. The reasoning graph is then linearized into a concrete sequence of sub-questions, creating a "golden" plan that serves an explicit supervision signal. Formally, the task decomposition process can be expressed as:

$$\mathcal{P} = \{q_1, q_2, \cdots, q_n\} = \left\{ f_{\text{TD}}(q_{<i}, q, a, \hat{\mathcal{F}}) \right\}_{i=1}^n. \qquad (3)$$

It should be noted that sub-questions in the initial plan may be under-specified. These incomplete questions will be dynamically grounded and refined during plan execution, as answers from preceding steps provide the necessary context (e.g., entities, constraints) for subsequent sub-questions.

### 4.1.2 ITERATIVE EXPLORATION

Upon obtaining the initial plan, EviPath generates the solution by iteratively simulating an agent's task problem-solving process. Each iteration consists of two primary steps: *think* and *action*.

**Think.** In this step, the planner of an RAG agent reviews answers to sub-questions resolved in preceding iterations $A_{i-1}$ and identifies the set of remaining sub-questions that are both solvable and essential to pursue in the current iteration. It then generates its internal monologue, or "thoughts" $\mathcal{R}_i$, enclosed within `<think>` and `</think>` tags. These thoughts detail: (i) the instantiation of previously underspecified variables, (ii) a prioritized set of sub-goals for the current step, and (iii) the resulting updates to the previous plan, and the dependencies among sub-questions. To maintain alignment with the target of the original question, the thought generation process is conditioned on the current agent state, $s_i = \left\{ \mathcal{P}, \{\mathcal{R}_j\}_{j=1}^{i-1}, \{Q_j\}_{j=1}^{i-1}, \{A_j\}_{j=1}^{i-1} \right\}$, along with the final answer $a$, and golden evidence set $\hat{\mathcal{F}}$. This step is formulated as:

$$\mathcal{R}_i = f_{\text{think}}(s_i, a, \hat{\mathcal{F}}). \tag{4}$$

**Action.** In this step, the planner translates the priorized sub-goal(s) from its thought into concrete, executable retrieval queries $Q_i$. Specifically, the retrieval intent is explicitly rendered within `<action>` and `</action>` tags. Each retrieval query is written as a complete sub-question $q_j \in Q_i$ that can be executed independently. Similarly, let $m_i$ to be the number of sub-questions needs to be solved in the $i$-th step, we have the objective function:

$$Q_i = \{q_1, q_2, \cdots, q_{m_i}\} = \left\{ f_{\text{action}}(\mathcal{R}_i, s_i, a, \hat{\mathcal{F}}) \right\}_{j=1}^{m_i}. \tag{5}$$

### 4.2 FAITHFUL SUB-QUESTION ANSWERING (EXECUTOR-SIDE REASONING PATH SYN.)

After the planning step generates a sub-question $q_i$, we synthesize the corresponding reasoning path for the executor. This involves generating a chain-of-thoughts $\mathcal{R}_i^{(s)}$ that processes the sub-question and its retrieved context to yield an intermediate answer $a_i$.

**The challenge of real-time retrieval.** In practice, the executor of a RAG agent retrieves the relevant context of the sub-question from an external knowledge base. However, existing dense or sparse retrievers often fail to secure the necessary golden evidence. The imperfect retrieval is particularly problematic since mainstream MHQA datasets do not provide intermediate answers at the sub-question level. Without such granular supervision, any disruption to the evidence chain prevents the LLM from assembling a coherent reasoning path and ultimately leading to incorrect or unfaithful answers.

**Robust trajectory synthesis in a simulated environment.** To circumvent the aforementioned challenge, EviPath forgoes real-time retrieval and instead constructs a simulated environment for robust data synthesis. By utilizing the complete set of supporting facts $\mathcal{F}$ as a stable, local knowledge base, we ensure all necessary golden evidence for each sub-question is readily accessible, creating an ideal setting for generating high-fidelity reasoning paths.

Within this simulated environment, we synthesize the reasoning path for each sub-question $q_i$ through a three-step procedure:

(i) First, we provide the LLM with the sub-question $q_i$ and the complete golden evidence set $\hat{\mathcal{F}}$ to **derive the intermediate answer**, $a_i$.

(ii) Then, we **identify the golden evidence set $\hat{\mathcal{F}}_i$ for the current sub-question** $q_i$. Specifically, we employ a sentence transformer to encode the complete answer sentence and every piece of golden evidence in $\hat{\mathcal{F}}$. All evidence having a cosine similarity to the answer that is higher than a threshold $\tau$ will be included in $\hat{\mathcal{F}}_i$. [1]

---

[1] If none of the evidence satisfies the condition, we pick the one with the highest cosine similarity.

(iii) Finally, following the abductive reasoning paradigm, we task the LLM to **generate a chain-of-thoughts** $\mathcal{R}_i^{(s)}$ that begins with sub-question $q_i$, identifies the set of key evidence $\hat{\mathcal{F}}_i$ from the noisy supporting facts $\mathcal{F}$, and culminates in the answer $a_i$. Considering that real-world retrieval results also contain substantial noise, leveraging $\mathcal{F}$ rather than $\hat{\mathcal{F}}$ as the pseudo retrieval context can better equip LLMs with better in-context reasoning ability.

Formally, the data synthesis process for answering sub-questions can be formulated as follows:

$$a_i = f_{\text{QA}}(q_i, \hat{\mathcal{F}}), \quad \hat{\mathcal{F}}_i = \{\zeta \in \hat{\mathcal{F}} \mid \cos(\phi(\zeta), \phi(a_i)) > \tau\}, \quad \mathcal{R}_i^{(s)} = f_{\text{think}}^{(s)}(q_i, a_i, \hat{\mathcal{F}}_i, \mathcal{F}), \quad (6)$$

where $\phi(\cdot)$ denotes the embedding encoded by sentence transformer $\phi$.

The reasoning path synthesis process continues until the final answer $a$ is reached. At this final step, the LLM refrains from issuing further `<action>` tags and concludes the trajectory by extracting the answer and wrapping it in `<answer>` and `</answer>` tags.

### 4.3 DATA FORMATTING AND CONVERSATIONAL FINE-TUNING

We generated **265k** process-supervised reasoning trajectories using the `LLaMA3.1-70B` model with few-shot demonstrations on the 2WikiMultihopQA (Ho et al., 2020), HotpotQA (Yang et al., 2018), and MuSiQue (Trivedi et al., 2022) training sets. [2] Each trajectory was then formatted to align with a RAG agent's architecture, yielding multi-turn *Planner Prompts* for training complex high-level planning capabilities, and single-turn *Executor Prompts* for training faithful, evidence-grounded sub-question answering.

The data from both prompt types are aggregated together in a unified supervised fine-tuning (SFT) process. Formally, the LLM is optimized by maximizing the following joint objective function:

$$\mathcal{J}_{\text{SFT}}(\theta) = \mathbb{E}_{(q,a,\mathcal{F}) \sim \mathcal{D}_{\text{train}}} \Big[ \pi_\theta(\mathcal{P} \mid q, \mathcal{I}_p) \cdot \prod_{i=1}^{|\mathcal{P}|-1} \pi_\theta(Q_i, \mathcal{R}_i \mid \mathcal{P}, Q_{<i}, a_{<i}, q, \mathcal{I}_p)$$

$$\cdot \Big( \prod_{j=1}^{|Q_i|} \pi_\theta(a_j, \mathcal{R}_j^{(s)} \mid q_j, \mathcal{F}, \mathcal{I}_e) \Big) \cdot \pi_\theta(a, \mathcal{R}_{|\mathcal{P}|} \mid \mathcal{P}, Q_{<|\mathcal{P}|-1}, a_{<|\mathcal{P}|-1}, q, a, \mathcal{I}_p) \Big], \quad (7)$$

where $\mathcal{D}_{\text{train}}$ denotes the training dataset, $\pi_\theta$ is the policy of the backbone LLM with trainable parameters $\theta$, $\mathcal{I}_p$ and $\mathcal{I}_e$ are instruction prompts for the planner and the executor, respectively. Detailed prompts for reasoning path synthesis and question answering are listed in Appendix G and H.

## 5 EXPERIMENTS

### 5.1 DATASETS, BASELINES AND EVALUATION METRICS

We conduct our main experiments on three multi-hop QA datasets, including text-based benchmarks HotpotQA (Yang et al., 2018) and MuSiQue (Trivedi et al., 2022) and knowledge graph-based question answering (KBQA) benchmark 2WikiMultihopQA (2Wiki) (Ho et al., 2020). We compare our proposed method EviPath with a comprehensive set of 24 baseline methods. We examine the performance of EviPath and all baseline methods with Exact Match (EM) and F1 scores. Details of datasets, baseline methods, and evaluation metrics are provided in Appendix B, C, and D.

### 5.2 IMPLEMENTATION

We examine the effectiveness of our proposed method by fine-tuning four instruction-tuned LLMs with different scales: `Qwen2.5-7B`, `LLaMA 3.2-1B`, `LLaMA 3.2-3B`, and `LLaMA 3.1-8B`.

During evaluation, we adopt the "**open-domain**" setting, where *the agent is required to retrieve relevant information from the external environment*, while disregarding the supporting facts provided with dev. set questions.

We adopt `bge-large-en-v1.5` as the retriever in all of our experiments. More details are in Appendix E.

---

[2]During training, trajectories from different datasets will never be mixed together.

| Method | Backbone | HotpotQA | | MuSiQue | | 2WikiMQA | | Average | |
|---|---|---|---|---|---|---|---|---|---|
| | | EM | F1 | EM | F1 | EM | F1 | EM | F1 |
| CoT* | GPT-4o | 29.4 | 48.9 | 17.0 | 28.9 | 41.8 | 53.6 | 29.4 | 43.8 |
| RAG* | GPT-4o | 47.2 | 63.6 | 17.4 | 30.1 | 45.8 | 57.1 | 36.8 | 50.3 |
| DecomP* | GPT-4o | 52.2 | 65.6 | 27.8 | 42.3 | 62.2 | 73.3 | 47.4 | 60.4 |
| RAFT | Llama3.1-8B | 41.0 | 51.6 | 13.8 | 24.0 | 39.4 | 45.8 | 31.4 | 40.5 |
| RaFe | GPT4o-mini | 40.6 | 55.4 | 12.4 | 25.3 | 36.2 | 39.3 | 29.7 | 40.0 |
| Iter-RetGen | Instruct-GPT | 45.1 | 60.4 | 26.1 | 42.0 | 50.2 | 65.3 | 40.5 | 55.9 |
| HippoRAG | GPT3.5 | 41.8 | 55.0 | 19.2 | 29.8 | 46.6 | 59.5 | 35.9 | 48.1 |
| IRCoT | GPT3 | 49.3 | 60.7 | 26.5 | 36.5 | 57.7 | 68.0 | 44.5 | 55.1 |
| RQ-RAG | GPT4o-mini | 46.4 | 59.4 | - | - | 50.2 | 58.8 | - | - |
| ReSP | Llama3-8B | 47.2 | - | - | - | 38.3 | - | - | - |
| IterDRAG | Gemini 1.5 | 38.4 | 49.8 | 22.6 | 35.0 | 44.3 | 54.6 | 35.1 | 46.5 |
| EfficientRAG | Llama3-8B | 50.6 | 57.9 | 16.4 | 21.2 | 44.2 | 51.6 | 37.1 | 43.6 |
| *RAG Agents* | | | | | | | | | |
| Search-o1 | QwQ 32B | 45.2 | 57.3 | 16.6 | 28.2 | 58.0 | 71.4 | 39.9 | 52.3 |
| RAG-Gym | Llama3.1-8B | 44.1 | 56.8 | - | - | 50.2 | 57.9 | - | - |
| Search-R1 | Qwen2.5-7B | 43.3 | - | 19.6 | - | 38.2 | - | 33.7 | - |
| R1-Searcher | Qwen2.5-7B | - | 60.4 | - | 35.7 | - | 62.8 | - | 53.0 |
| Collab-RAG | Llama3.1-8B | 53.0 | 65.6 | 26.4 | 42.4 | 63.2 | 74.6 | 47.5 | 60.9 |
| RAG-Star | GPT4o-mini | 46.0 | 60.0 | 22.2 | 30.7 | 38.0 | 46.8 | 35.4 | 45.8 |
| Mujica-MyGO | Qwen2.5-7B | 41.5 | 53.8 | 26.1 | 35.9 | 77.6 | 84.2 | 48.4 | 58.0 |
| *Concurrent Works* | | | | | | | | | |
| R1-Searcher++ | Qwen2.5-7B | - | 59.0 | - | 33.8 | - | 61.2 | - | 51.3 |
| DynaSearcher | Qwen2.5-7B | 52.0 | 66.1 | 26.5 | 38.7 | 61.9 | 72.0 | 46.8 | 58.9 |
| KG-o1 | Llama3.1-8B | 43.4 | 60.2 | - | - | 55.0 | 68.6 | - | - |
| ESA-KGR | Qwen2.5-7B | 36.8 | 47.3 | 10.5 | 18.0 | 49.5 | 58.1 | 32.3 | 41.1 |
| Graph-R1 | Qwen2.5-7B | - | 62.7 | - | 46.2 | - | 65.0 | - | 58.0 |
| ***Ablation Experimental Results of* EviPath** | | | | | | | | | |
| - w/ pretrained LLM | Llama3.1-8B | 19.5 | 30.8 | 6.2 | 14.7 | 57.6 | 62.3 | 27.8 | 35.9 |
| - w/ pretrained LLM | Llama3.1-70B | 31.0 | 44.9 | 13.1 | 23.4 | 84.6 | 87.7 | 42.9 | 52.0 |
| - w/o planner fine-tuning | Llama3.1-8B | 41.6 | 54.3 | 27.6 | 37.7 | 45.2 | 50.4 | 38.1 | 47.5 |
| - w/o executor fine-tuning | Llama3.1-8B | 48.9 | 61.8 | 31.2 | 41.9 | 86.1 | 91.6 | 55.4 | 65.1 |
| - w/o supporting facts | Llama3.1-8B | 51.8 | 65.0 | 34.0 | 44.3 | 91.1 | 93.4 | 59.0 | 67.6 |
| **EviPath (full)** | Qwen2.5-7B | 51.3 | 64.0 | 40.2 | 50.0 | **92.0** | **94.3** | 61.2 | 69.4 |
| | Llama3.2-1B | 39.4 | 50.6 | 29.7 | 37.9 | 76.7 | 79.0 | 48.6 | 55.8 |
| | Llama3.2-3B | 48.6 | 60.7 | 39.9 | 48.8 | 90.4 | 92.9 | 59.6 | 67.4 |
| | Llama3.1-8B | **53.8** | **66.4** | **44.3** | **54.6** | 91.3 | 93.6 | **63.1** | **71.5** |

Table 1: Experiment results on multi-hop question answering benchmark datasets. The performance of vanilla CoT, RAG, and DecomP (with *) are referred from Xu et al. (2025). Results for other baselines are taken from original research papers.

## 5.3 MAIN RESULTS

The experiment results in Table 1 demonstrate that EviPath is a simple yet effective scheme for synthesizing reasoning trajectories for training RAG agents. Despite relying solely on SFT, our 8B model trained on EviPath-synthesized trajectories significantly outperforms all baselines, including those leveraging large-scale LLMs (e.g. GPT-4o) or complex RL algorithms (e.g. GRPO), achieving an average absolute EM gain of **14.7%**. The substantial improvement reaffirms persistent data limitations in RAG agent training and highlights the importance of introducing precise, evidence-anchored reasoning paths. Our EviPath-trained agents exhibit a clear scaling effect, with larger backbone LLMs consistently improving QA performance. ***More importantly, our results demonstrate that the process-supervised trajectories can offset model sizes, enabling smaller LLMs to overcome their limited reasoning capabilities.*** Specifically, RAG agents equipped with 1B and 3B `LLaMA 3.2` models trained on EviPath data achieve state-of-the-art performance on the 2WikiMQA and MuSiQue datasets and substantially surpass all baseline models.

EviPath excels on both text-based and knowledge-based multi-hop QA. Its strong performance on 2WikiMQA highlights its ability to leverage knowledge graphs, whose structured nature helps capture the logical dependencies between sub-questions. Unlike other KG-based baselines (e.g., Graph-R1, KG-o1), EviPath's evidence-anchored process supervision compels RAG agents to remain faithful to the graph structure, and hence, encourages the selection of optimal reasoning paths.

| Datasets: (Training - Eval.) | | HotpotQA - MuSiQue | | MuSiQue - HotpotQA | | 2WikiMQA - QALD10 | |
|---|---|---|---|---|---|---|---|
| Method | Backbone | EM | F1 | EM | F1 | EM | F1 |
| Search-R1 | Qwen2.5-3B | - | 5.03 | - | 19.8 | - | - |
| Graph-R1 | Qwen2.5-3B | - | 33.1 | - | 49.8 | - | - |
| **EviPath (Ours)** | Llama3.2-3B | 30.2 | 39.2 | 34.6 | 44.8 | 40.1 | 45.4 |
| Mujica-MyGO | Qwen2.5-7B | 26.1 | 35.9 | - | - | 39.9 | **49.7** |
| **EviPath (Ours)** | Llama3.1-8B | **35.9** | **46.3** | **38.8** | **50.1** | **43.9** | 48.6 |

Table 2: Experimental results in out-of-domain settings.

| Method | Data Synthesis LLM | RAG Agent Backbone | HotpotQA | | MuSiQue | | 2WikiMQA | | Average | |
|---|---|---|---|---|---|---|---|---|---|---|
| | | | EM | F1 | EM | F1 | EM | F1 | EM | F1 |
| **EviPath (Ours)** | Llama3.1-8B | Llama3.1-8B | 50.9 | 63.5 | 39.1 | 49.0 | 86.1 | 90.3 | 58.7 | 67.6 |
| **EviPath (Ours)** | Llama3.1-70B | Llama3.1-8B | **53.8** | **66.4** | **44.3** | **54.6** | **91.3** | **93.6** | **63.1** | **71.5** |

Table 3: Question answering performance of RAG agents trained with reasoning paths synthesized by different LLMs.

The performance gain on the HotpotQA dataset is relatively modest, which can be attributed to its lower complexity. Since HotpotQA only consists two-hop questions, it demands less on the inherent reasoning capabilities of LLMs, thus narrowing the gap among all methods.

The performance gain on the HotpotQA dataset is relatively modest, which can be attributed to its lower complexity. Since HotpotQA only consists two-hop questions, it demands less on the inherent reasoning capabilities of LLMs, thus narrowing the gap among all methods. In contrast, EviPath's advantage is pronounced on the more complex MuSiQue dataset with various types of 2-to-4 hops questions, especially over policy gradient optimization-based baselines (e.g. Search-R1). This highlights a key limitation of RL: without a foundational ability to solve a problem, an agent cannot acquire the positive rewards needed for self-improvement. ***To sum, our findings suggest that the primary bottleneck in training powerful question answering agents may not be the learning algorithm or model scale, but the availability of high-quality, process-level supervision signal.***

## 5.4 ABLATION STUDIES

We examine the effectiveness of the proposed EviPath pipeline in different settings by answering the following research questions (RQs).

**RQ1: Does the use of question-specific supporting evidence improve the quality of synthesized reasoning paths?** We evaluated the necessity of supporting evidence by reconfiguring our data synthesis pipeline to use only question–answer pairs, compelling the LLM to retrieve relevant contexts from an external knowledge base and construct a complete reasoning path. As detailed in Table 1, the exclusion of supporting evidence resulted in performance degradation for our 8B LLM-based RAG agent across all three datasets. This performance drop is attributable to the loss of the implicit reasoning path implied in supporting evidence, which typically constrains the model's search space and ensures faithful derivations. In its absence, the model is vulnerable to two failure modes: (1) imperfect retrieval, where the inability to find "golden" evidence leads to plausible but incorrect reasoning, and (2) inherent limitations in the LLM's ability to reason about complex questions without explicit guidance. The latter issue is particularly acute on MuSiQue, which demands the composition of multiple facts and thus exhibits the most severe degradation.

**RQ2: Which core capability of LLMs is the primary limitation for building RAG agents?** To identify the primary limitation of LLMs in RAG agents, we trained two specialized models: a planner for high-level planning and an executor for sub-question answering. As shown in Table 1, replacing either specialized model with a pre-trained LLM degrades performance. Notably, this degradation is far more pronounced when replacing the planner. This result indicates that the primary bottleneck of LLM is not semantic understanding but long-horizon planning and reasoning, which reaffirms the critical need for high-quality reasoning trajectories in RAG agent development.

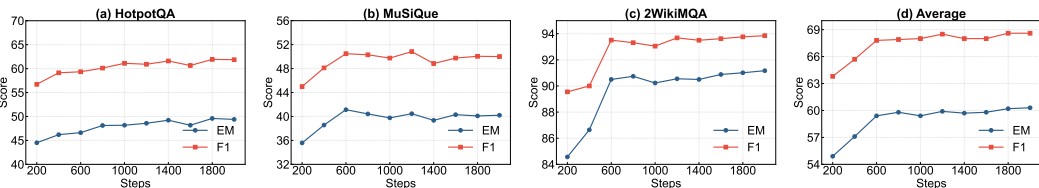

Figure 5: Step-wise EM/F1 score on three open-domain QA datasets based on Qwen 2.5-7B model.

**RQ3: Can RAG agents trained on EviPath-synthesized reasoning paths generalize to out-of-domain scenarios?** We evaluate out-of-domain (OOD) generalization using cross-dataset transfer from HotpotQA to MuSiQue and vice versa (for textual QA), as well as from 2Wiki-KG to QALD-10 (for KBQA). [3] As shown in Table 2, models fine-tuned on EviPath-synthesized trajectories achieve comparable or superior transferability to state-of-the-art methods optimized with GRPO (Shao et al., 2024). Most notably, a model fine-tuned exclusively on 2-hop questions from HotpotQA shows remarkable generalization: it not only surpasses OOD baselines on MuSiQue (which features more complex 3–4 hop questions) but also outperforms all in-domain baselines trained directly on the training subset of MuSiQue.

**RQ4: To what extent does imperfect retrieval affect the end-to-end QA performance?**

To isolate the impact of retrieval accuracy on end-to-end QA performance, we also evaluate EviPath-trained RAG agents in the distractor setting, where we use the 20 supporting facts (paragraphs) provided with each test sample to simulate the retrieval results. From Figure 4 we observe that when the golden evidence is guaranteed to be included in the retrieval results, the performance ceiling rises substantially, underscoring the need for developing more advanced retrieval methods. In addition, the distractor setting allows us to make a direct comparison between EviPath and state-of-the-art reasoning path synthesis baselines, namely LongAlphaca (Chen et al., 2024), LongAlign (Bai et al., 2024), LongReward (Zhang et al., 2025a),

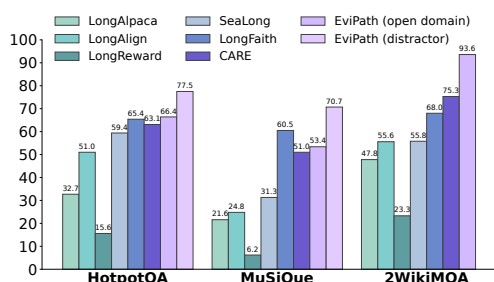

Figure 4: End-to-end QA performance (in F1) with LLaMA-3.1-8B model under the distractor setting (N/A for KBQA).

SeaLong (Li et al., 2024), LongFaith (Yang et al., 2025), and CARE (Wang et al., 2025b). The double-digit gains demonstrate that while agentic RAG systems are not explicitly designed for the distractor setting, they still outperform non-agentic LLMs in long-context reasoning.

**RQ5: How does the capacity of reasoning path generator affect the quality of synthetic reasoning paths and the final question answering performance?** To investigate the impact of the underlying LLM's capacity on the quality of EviPath synthesized trajectories, we generated a new set of training data using the Llama-3.1-8B model. From Table 3 we conclude that RAG agents trained on reasoning paths synthesized by the 70B Llama model yield better QA performance. However, an 8B model is already sufficient to synthesize high-quality reasoning paths that allow a RAG agent to attain state-of-the-art performance, demonstrating the robustness of the EviPath paradigm.

**RQ6: How does increasing training data gradually improve QA performance?** Figure 5 shows that model performance scales with the volume of training data during the early stages of the SFT process, but the gains exhibit diminishing returns. Moreover, on large datasets like HotpotQA and 2Wiki, training for 2,000 steps (approx. 20,000 examples) achieves performance within approximately 1% of that from training on the full dataset. This represents a significant efficiency advantage over policy gradient optimization, as it avoids the need for repetitive rollout exploration.

**RQ7: To what extent does abductive reasoning improve the quality of synthetic reasoning paths?** To isolate the benefit of abductive reasoning, we re-synthesize the reasoning paths using a deductive, self-generation approach. In this setup, a Llama-3.1-8B model-based RAG agent is instructed to answer training questions without access to ground-truth answers or supporting facts. The results in Table 4 show that SFT offers limited improvement when it primarily reinforces skills and

---

[3]QALD-10 does not have a training set.

| Method | Data Synthesis LLM | Data Synthesis Mode | RAG Agent Backbone | HotpotQA | | 2WikiMQA | |
|---|---|---|---|---|---|---|---|
| | | | | EM | F1 | EM | F1 |
| EviPath | N/A | N/A | Llama3.1-8B (pt.) | 19.5 | 30.8 | 57.6 | 62.3 |
| EviPath | Llama3.1-8B (pt.) | Deductive | Llama3.1-8B (ft.) | 39.7 | 51.2 | 80.3 | 85.7 |
| EviPath | Llama3.1-8B (pt.) | Abductive | Llama3.1-8B (ft.) | **50.9** | **63.5** | **86.1** | **90.3** |

Table 4: Performance comparison between deductive and abductive reasoning path synthesis. Here, "pt." and "ft." stand for pre-trained and fine-tuned models, respectively.

| Method | Deployment | Backbone | HotpotQA | | MuSiQue | | 2WikiMQA | | Average | |
|---|---|---|---|---|---|---|---|---|---|---|
| | | | EM | F1 | EM | F1 | EM | F1 | EM | F1 |
| EviPath | Multiple LLMs | 2×Llama3.1-8B | 53.3 | 65.9 | 43.6 | 53.4 | 90.2 | 92.6 | 62.4 | 70.6 |
| EviPath | Single LLM | Llama3.1-8B | **53.8** | **66.4** | **44.3** | **54.6** | **91.3** | **93.6** | **62.8** | **71.3** |

Table 5: Question answering performance comparison between different LLM deployment settings.

knowledge that the LLM already possesses. In contrast, our abductive approach reverse-engineers paths from answers and supporting facts, which effectively lowers the dependency on model priors, unlocking a significantly higher performance ceiling.

**RQ8: Does deploying the Planner and the Executor modules of a RAG agent to two specialized LLMs outperform the same RAG agent supported by one single LLM?** The planner and executor modules address distinct aspects of the question answering task. To compare deployment strategies, we trained both a single LLM on all data and two specialized LLM on partitioned data. As shown in Table 5, the performance difference between the single-LLM and dual-LLM setups is modest. The slight advantage of the single-LLM setup suggests a positive transfer learning effect, where training on both planning and in-context reasoning tasks is mutually beneficial. This indicates that the key to unlocking the potential of small-scale LLMs is not specialization, but the quality of the training data.

## 5.5 CASE STUDY

**RQ9: Are RAG agents trained by synthetic reasoning trajectories robust enough when encountering unexpected situations (e.g. incomplete retrieval and failed plan)?** In real-world scenarios, RAG agents may receive negative feedback from the external environment. Specifically, we encounter incomplete retrieval results in $978$ out of the $7,405$ questions in the HotpotQA development set. However, our RAG agents is still able to correctly answer 291 of these questions, achieving an accuracy of $29.8\%$, which demonstrates strong robustness. From examining the actual trajectory outputs, we observe that our RAG agent primarily employs two strategies to cope with incomplete or failed retrieval: (i) rephrasing the sub-question and issuing an additional search query (see Table 14 and 15 in Appendix I for details), and (ii) making inferences or educated guesses based on its internal knowledge (see Table 15). In addition, actual trajectory outputs also show that the RAG agent trained by EviPath synthesized data is also able to conduct self-reflection and plan revision. For detailed case illustration and analysis, please refer to Table 16 in Appendix I.

## 6 CONCLUSION

In this paper, we introduced EviPath, a novel framework that uniquely applies abductive reasoning to reverse-engineer complete, evidence-anchored reasoning paths that include explicit task decomposition, retriever use, reasoning thoughts, and intermediate answers. EviPath overcomes the fundamental limitations of outcome-rewarded RL and static CoT trajectory synthesis methods, establishing an efficient, data-centric paradigm for RAG agent development. Experiments on commonly adopted open-domain QA benchmarks demonstrate that EviPath-synthesized data significantly boosts in-domain accuracy and out-of-domain generalization of RAG agents. In the future, we plan to investigate the integration of process-supervised signals with policy gradient optimization methods and explore the potential of extending our data synthesis paradigm to other agentic tasks.

ETHICS STATEMENT

To the best of our knowledge, this work does not involve any discrimination, social bias, or private data. All reasoning paths are constructed from open-source datasets and knowledge bases, namely Wikidata and Wikipedia. Therefore, we believe that our study complies with the Ethics Policy of the ICLR conference.

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

# A EXTENDED RELATED WORKS

## A.1 RAG AGENTS FOR QA

RAG agents couple large language models (LLMs) with iterative search to solve knowledge-intensive question answering (QA). While modern LLMs exhibit strong reasoning capabilities, they still suffer from knowledge hallucinations, motivating the use of external evidence via retrieval (Achiam et al., 2023; Cheng et al., 2024; Huang et al., 2025; Gao et al., 2023). Two integration patterns dominate: (i) *pipeline RAG*, which retrieves once (or in a few rounds) and feeds the concatenated passages to the LLMs (Lewis et al., 2020; Gao et al., 2023), and (ii) *tool-augmented retrieval*, where the model plans, calls tools, and verifies in a closed loop. The former is simple yet often under-recovers multi-hop evidence, a common failure mode in open-domain QA benchmarks such as HotpotQA (Yang et al., 2018), 2WikiMultiHopQA (Ho et al., 2020), and MuSiQue (Trivedi et al., 2022). The latter explicitly interleaves reasoning with retrieval and tool use: ReAct (Yao et al., 2023) and IRCoT (Trivedi et al., 2023) guide iterative search with chain-of-thought reasoning; Schick et al. (2023) imparts API/tool usage via supervised fine-tuning; SelfRAG (Asai et al., 2024) and Iter-RetGen (Shao et al., 2023) include reflection into the RAG synergy. Recent approaches further scale agentic QA through learned inner monologues and process signals, IM-RAG introduces mid-step rewards for multi-round search (Yang et al., 2024), RAG-Gym provides a process-supervised environment for optimizing search agents (Xiong et al., 2025), Collab-RAG coordinates white-/black-box LLMs for complex QA (Xu et al., 2025), while RAG-Star augments tree-style deliberation with retrieval-aware verification (Jiang et al., 2025a). In parallel, Search-o1 (Li et al., 2025) decomposes the high-level planning and low-level in-document reasoning into different modules. Recently, Search-R1 (Jin et al., 2025), R1-searcher Song et al. (2025a), and Mujica-MyGO (Wu et al., 2025) propose leveraging policy gradient optimization approaches to improve the reasoning capability of QA agents. Orthogonal advances strengthen the retrieval side, including query reformulation for RAG system (Chan et al., 2024; Mao et al., 2024) and end-to-end multi-hop retrievers that maintain passage hypotheses across hops (Zhang et al., 2024a). Graph-R1 (Luo et al., 2025), KG-o1 (Wang et al., 2025a) and DynaSearcher (Hao et al., 2025) further utilize knowledge graphs to improve the precision and relevance of retrieved contexts. Collectively, agentic RAG for QA has progressed from one-shot "retrieve-and-read" to interactive planning, tool-use, and verification. However, existing state-of-the-art approaches still rely on outcome rewards (Song et al., 2025a; Wu et al., 2025; Hao et al., 2025) or prompt-level heuristics (Li et al., 2025) and seldom expose *evidence-anchored, stepwise* trajectories that align question decomposition, tool invocations, and intermediate verification precisely with the training signal our work targets.

## A.2 REASONING PATH SYNTHESIS

Enhancing the reasoning capabilities of LLMs has garnered significant attention, driving the development of data synthesis methods (Wang et al., 2023; Xiong et al., 2024; Bai et al., 2024; Yang et al., 2025). Earlier approaches like GENREAD (Yu et al., 2023) and SP-CoT (Wang et al., 2023) focus on replacing retrieval with model-generated retrieval, but the synthesized reasoning paths are not grounded in evidence and thus remain vulnerable to hallucination. More recent works line in improving the long-context processing capabilities of LLMs by constructing continued pretraining data Xiong et al. (2024); Gao et al. (2025), concatenating context segments into long training sequences to address the lost-in-the-middle problem (An et al., 2024), or generating step-wise CoT trajectories to answer complex, multi-hop questions (Bai et al., 2024). In order to make the reasoning trajectories faithful and grounded, LongFaith (Yang et al., 2025) proposes to include reasoning thoughts with chains of golden evidence citations, which effectively alleviates hallucinations and achieves desirable results. Nevertheless, these approaches focus on generating the chain-of-thought reasoning steps based on the fixed contexts, failing to guide the training of RAG agents that necessitate environment interactions.

# B DATASET STATISTICS

We conduct our main experiments on three multi-hop QA datasets, including text-based benchmarks HotpotQA (Yang et al., 2018) and MuSiQue (Trivedi et al., 2022), and KBQA benchmark 2WikiMultihopQA (Ho et al., 2020). In our ablation studies, we also include another KBQA dataset

| Datasets | #Train | #Dev | #Test | #Hops | Corpus |
|----------|--------|------|-------|-------|--------|
| HotpotQA | 90447 | 7405 | 7405 | 2 | Text |
| MuSiQue | 19938 | 2417 | 2459 | 2-4 | Text |
| 2WikiMultihopQA | 167454 | 12576 | 12576 | 2-5 | Text and KG |
| QALD-10 | - | 394 | - | 1-2 | KG |

Table 6: Statistics of Datasets

QALD-10 (Usbeck et al., 2024) for out-of-domain evaluation. Table 6 shows the detailed statistics for all four datasets.

## C  BASELINES

We compare our proposed method EviPath with a comprehensive set of 24 baseline methods, including vanilla CoT (Wei et al., 2022), and RAG (Lewis et al., 2020), iterative RAG pipelines such as DecomP (Khot et al., 2023), RAFT (Zhang et al., 2024b), RaFe (Mao et al., 2024), Iter-RetGen (Shao et al., 2023), HippoRAG (Gutiérrez et al., 2024), IRCoT Trivedi et al. (2023), RQ-RAG (Chan et al., 2024), ReSP (Jiang et al., 2025b), IterDRAG (Yue et al., 2025) and EfficientRAG (Zhuang et al., 2024), RAG agents such as Search-o1 (Li et al., 2025), RAG-Gym Xiong et al. (2025), Search-R1 (Li et al., 2025), Collab-RAG (Xu et al., 2025), RAG-Star Jiang et al. (2025a), R1-Searcher (Song et al., 2025a), Mujica-MyGO (Wu et al., 2025) and concurrent works such as R1-Searcher++ (Song et al., 2025b), DynaSearcher (Hao et al., 2025), KG-o1 (Wang et al., 2025a), ESA-KGR (Zhang et al., 2025b), and Graph-R1 (Luo et al., 2025).

## D  EVALUATION METRICS

We examine the performance of EviPath and all baseline methods using the Exact Match (EM) ratio and token-level F1 scores. EM measures the fraction of predicted answers $\hat{y}$ that are identical to ground truth answers $y$ after normalization. For a development set $\mathcal{D}_{\text{dev}}$, the EM ratio is calculated as:

$$\text{EM} = \sum_{i=1}^{|\mathcal{D}_{\text{dev}}|} \mathbb{1}\big(\text{norm}(\hat{y}_i) = \text{norm}(y_i)\big), \tag{8}$$

where $\mathbb{1}(\cdot)$ is the indicator function. The $\text{norm}(\cdot)$ function lowercases text and removes articles, punctuation, and leading/trailing spaces. Similarly, we can calculate the F1 ratio as follows:

$$F_1 = \sum_{i=1}^{|\mathcal{D}_{\text{dev}}|} \frac{2 \cdot p_i \cdot r_i}{p_i + r_i}, \tag{9}$$

$$\text{where } p_i = \frac{|T(\text{norm}(\hat{y}_i)) \cap T(\text{norm}(y_i))|}{|T(\text{norm}(\hat{y}_i))|},$$

$$r_i = \frac{|T(\text{norm}(\hat{y}_i)) \cap T(\text{norm}(y_i))|}{|T(\text{norm}(y_i))|}.$$

Here, $p_i$ and $r_i$ denote the token-level precision and token-level recall for the $i$-th development set question, respectively. $T(\cdot)$ denotes the tokenization process.

## E  IMPLEMENTATION DETAILS

We examine the effectiveness of our proposed method by fine-tuning four instruction-tuned LLMs with different scales: `Qwen2.5-7B`, `LLaMA 3.2-1B`, `LLaMA 3.2-3B`, and `LLaMA 3.1-8B`. During the training process, ***trajectories from different datasets are never mixed together***, which eliminates the possibility of cross-dataset data leakage. The number of training trajectories adopted for the HotpotQA, MuSiQue, and 2wiki datasets are about 86k, 20k, and 159k, respectively.

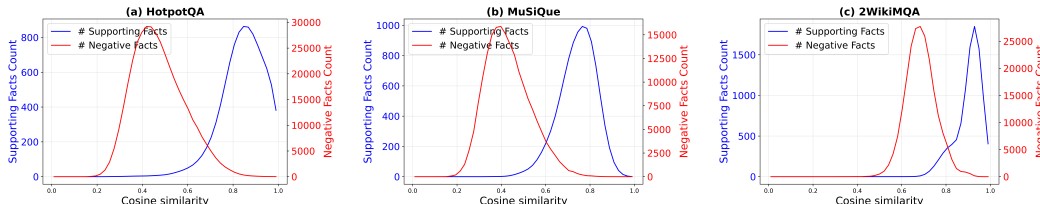

Figure 6: Cosine similarity distribution between sub-question answers and supporting/negative facts.

During evaluation, we adopt the "***open-domain***" setting, where the agent is required to retrieve relevant information from the external environment. Therefore, similar to all baseline methods, RAG agents developed with EviPath synthesized data also need to interact with "real environment", and handle noisy or incomplete retrieval results.

For the HotpotQA dataset, we adopt the official Wikipedia dump associated to the dataset as our retrieval corpus. For MuSiQue, we form a large retrieval corpus by aggregating all supporting passages associated with each question following the convention of baseline methods (Trivedi et al., 2023). As for 2Wiki and QALD-10, we utilize the official Web APIs provided by Wikidata (Vrandečić & Krötzsch, 2014). We adopt `bge-large-en-v1.5` as the retriever in all of our experiments.

All of our experiments are conducted on 4 GPUs, each equipped with 80GB VRAM. We leverage the vLLM framework for accelerated inference during reasoning path synthesis, using a tensor parallel size of $4$. We train all backbone LLMs with full-parameter fine-tuning using LLaMA-Factory (Zheng et al., 2024). Specific hyperparameters are detailed in Table 7.

| Hyperparameters | Settings |
|---|---|
| Threshold $\tau$ | 0.9 |
| SFT learning rate | 2e-6 |
| Per-device batch size | 2 |
| Gradient accumulation step | 8 |
| # Epochs | 2 |
| Warmup ratio | 0.1 |
| Top-p | 0.8 |
| LLM inference temperature | 0.7 |
| Max. output tokens | 4096 |
| Repetition penalty | 1.05 |

Table 7: Hyperparameter settings.

**Sensitivity analysis for evidence selection threshold $\tau$.** In order to analyze the sensitivity of evidence selection threshold $\tau$, we select 10000 sub-questions from the three datasets and calculate the cosine similarity between their answers and the supporting (golden) or negative facts. Figure 6 shows the overall distribution.

It should be mentioned that ***multi-hop QA datasets do not provide official question decomposition, answers for sub-questions, or ground truth evidence annotations for each of the sub-questions***. Therefore, we extract the evidence and sub-question answers from our synthetic trajectories. ***The evidences mentioned between*** `<select>` ***and*** `</select>` ***tags are considered as supporting facts, while others are considered as negative facts.***

From Figure 6, we can conclude that most supporting facts have a cosine similarity of around $0.9$ with the sub-question answers. In the MuSiQue dataset, the similarities between both supporting facts and negative facts and the answers are generally lower, whereas in the 2WikiMQA dataset they are overall higher. Therefore, selecting $0.9$ as the similarity threshold appears to be a reasonable choice.

We observe that some sub-questions have a low overall similarity, resulting in no evidence satisfying the threshold condition. To handle such cases, we select the evidence with the highest cosine similarity as the "golden evidence". Based on this grounding strategy, we compute the precision,

| Threshold $\tau$ | HotpotQA | | | MuSiQue | | | 2WikiMQA | | |
|---|---|---|---|---|---|---|---|---|---|
| | Prec. | Rec. | F1 | Prec. | Rec. | F1 | Prec. | Rec. | F1 |
| 0.6 | 0.276 | 0.987 | 0.431 | 0.688 | 0.970 | 0.805 | 0.077 | 0.999 | 0.143 |
| 0.7 | 0.603 | 0.961 | 0.741 | 0.884 | 0.936 | 0.909 | 0.256 | 0.999 | 0.408 |
| 0.8 | 0.854 | 0.915 | 0.883 | 0.914 | 0.916 | 0.914 | 0.710 | 0.968 | 0.819 |
| 0.9 | 0.895 | 0.890 | 0.892 | 0.916 | 0.915 | 0.915 | 0.934 | 0.958 | 0.946 |
| 0.95 | 0.896 | 0.888 | 0.892 | 0.916 | 0.915 | 0.915 | 0.958 | 0.952 | 0.955 |

Table 8: The effect of adjusting the threshold $\tau$ on evidence selection precision, recall, and F1.

| Method (Backbone / Setting) | HotpotQA | | MuSiQue | | 2WikiMQA | | Average | |
|---|---|---|---|---|---|---|---|---|
| | EM | F1 | EM | F1 | EM | F1 | EM | F1 |
| IRCoT (GPT3)* | 49.3 | 60.7 | 26.5 | 36.5 | 57.7 | 68.0 | 44.5 | 55.1 |
| IRCoT (GPT-4o) | 49.0 | 64.3 | 31.2 | 45.4 | 58.0 | 71.9 | 46.1 | 60.5 |
| HippoRAG (GPT3.5)* | 41.8 | 55.0 | 19.2 | 29.8 | 46.6 | 59.5 | 35.9 | 48.1 |
| HippoRAG (GPT-4o) | 43.7 | 57.7 | 26.9 | 39.0 | 59.4 | 67.9 | 43.3 | 54.9 |
| IRCoT+HippoRAG (GPT3.5)* | 45.7 | 59.2 | 21.9 | 33.3 | 47.7 | 62.7 | 38.4 | 51.7 |
| IRCoT+HippoRAG (GPT-4o) | 49.2 | 64.0 | 31.5 | 44.2 | 68.0 | 77.2 | 49.6 | 61.8 |
| **EviPath (w/o Golden Evid., Llama3.1-8B)** | 51.8 | 65.0 | 34.0 | 44.3 | 91.1 | 93.4 | 59.0 | 67.6 |
| **EviPath (Llama3.2-3B)** | 48.6 | 60.7 | 39.9 | 48.8 | 90.4 | 92.9 | 59.6 | 67.4 |
| **EviPath (Llama3.1-8B)** | **53.8** | **66.4** | **44.3** | **54.6** | **91.3** | **93.6** | **63.1** | **71.5** |

Table 9: Comparison between EviPath and Pipeline RAG-based question answering solution. Results with * are referred from original research papers. Other results of baseline methods are reproduced with the publicly available code.

recall, and F1 score of evidence selection under different thresholds. The results in Table 8 further support the choice of $0.9$ as the threshold. Lowering this threshold reduces the precision of labeling supporting facts for sub-questions. For the 2WikiMQA dataset, increasing the threshold beyond $0.9$ could potentially yield higher evidence-labeling precision. However, given the strong performance of the RAG agent on this dataset (EM $> 90\%$), we adopt a more conservative threshold to remain consistent with other datasets.

# F  DISCUSSION

**RQ10: Are existing state-of-the-art commercial LLMs capable of mastering complex question answering with specified RAG pipelines or workflows?** In order to answer this question, we re-examine two key state-of-the-art pipeline RAG methods, namely IRCoT (Trivedi et al., 2023) and HippoRAG (Gutiérrez et al., 2024) with the state-of-the-art commercial LLM: GPT-4o. From the experimental results in Table 9, we observe that replacing the backbone model with the more advanced GPT-4o leads to some performance improvements in pipeline RAG systems. However, these gains are insufficient to surpass the RAG agent developed on synthetic data produced by EviPath. Despite benefiting from the combined strengths of IRCoT's chain-based retrieval and HippoRAG's memory indexing, GPT-4o still lags behind the EviPath-trained 8B RAG agent by $13.5\%$ in EM.

Hence, we can conclude that: (i) pipeline RAG systems rely intensively on the capacity of backbone models, which often requires ultra-large LLMs like GPT3-175B, GPT3.5, GPT-4o to deliver adequate performance. (ii) In contrast, EviPath-synthesized trajectories empower small-scale LLMs to surpass their capability ceilings, yielding significant improvements in complex question answering.

**RQ11: RL algorithms can enable LLMs to self-explore trajectories relying solely on the final reward and engage in self-reflection and correction, is it necessary to synthesize a large amount of trajectory data in this scenario?** Yes, we believe that data curation and the synthesis of high-quality reasoning trajectories remain crucial, even in the era of RL-based methods. There are three main reasons:

**(1) Existing LLMs are still far from solving complex problems reliably.**

While modern LLMs have much stronger foundational abilities than earlier models, their multi-hop QA performance is still not sufficiently desirable. For example, we evaluated the pre-trained Llama-3.1-70B model on multi-hop QA (see Table 1), and its performance on MuSiQue remains very low (EM = 13.1), suggesting that the agentic multi-hop reasoning capability is still limited.

**(2) Performing SFT on synthetic trajectories is more efficient than outcome-rewarded RL.**

RL methods such as GRPO (Shao et al., 2024), GSPO (Zheng et al., 2025) and DAPO (Yu et al., 2025) typically require repeated rollouts and multiple trajectories per question to obtain accurate advantage estimates, which makes training costly. In contrast, SFT on EviPath's synthetic trajectories eliminates the need for repeated online sampling, and directly provides supervised signals for intermediate steps.

**(3) RL is limited by the LLM's inherent capability ceiling.**

Specifically, if an LLM is intrinsically unable to solve a certain class of problems, it will struggle to obtain meaningful positive rewards through self-exploration alone. In such regimes, RL may be ineffective because the LLM rarely, if ever, discovers successful trajectories from which to learn. In these cases, guidance from gold evidence and gold trajectories is crucial: it can bootstrap the model into regimes where RL becomes viable, or even provide capabilities that RL alone would not easily acquire.

In summary, despite the progress of strong LLM backbones and RL algorithms, high-quality synthetic reasoning trajectories remain necessary and complementary: they improve sample efficiency, reduce reliance on expensive rollouts, and help overcome the inherent limitations of purely reward-driven self-exploration.

# G    PROMPT TEMPLATES OF DATA SYNTHESIS

Table 10: Data synthesis prompt template for the planner module

---

**## System Prompts ##**

---

Your task is to provide a reasoning path to answer a complex question.
You will be given a question, the answer of the question, and supporting evidence to obtain the answer.
You need to:
1. Make a plan about how to answer this question by decompose it into multiple steps.
2. Solve the problem in a step-by-step manner.
Put your thoughts between `<think>` and `</think>` tags.
You can use the `Search(entity, "question")` function to get information.
Extract candidate entity from the intermediate subquestions with `[X]`.
Label the final answer with `<answer> *** </answer>`.

---

**## User ##**

Question: Which film has the director born earlier, Ek Paheli or When Father Was Away On Business?
Answer: Ek Paheli
Supporting evidences:
(Ek Paheli, director, Naresh Kumar)
(When Father Was Away on Business, director, Emir Kusturica)
(Naresh Kumar (tennis), date of birth, 22 December 1928)
(Emir Kusturica, date of birth, 24 November 1954)

Question: Which film has the director born earlier, Ek Paheli or When Father Was Away On Business? Candidate: [0] Ek Paheli, [1] When Father Was Away On Business
Answer: Ek Paheli

---

**## Assistant ##**

`<think>`
To solve this problem, I need to:
1. Find the director of film Ek Paheli.
2. Find the director of film When Father Was Away On Business.
3. Figure out the date of birth of the director of Ek Paheli.
4. Figure out the date of birth of the director of When Father Was Away On Business.
5. Compare their dates of birth to determine which director was born earlier.
`</think>`
`<action>`
Search([0], "Who is the director of Ek Paheli?")
Search([1], "Who is the director of When Father Was Away On Business?")
`</action>`

---

**## User ##**

Observation:
The director of Ek Paheli is Naresh Kumar. Candidate: [2] Naresh Kumar
The director of When Father Was Away On Business is Emir Kusturica. Candidate: [3] Emir Kusturica

---

**## Assistant ##**

`<think>`
Now that I have the directors of both films, I need to find their dates of birth.
`</think>`
`<action>`
Search([2], "What is the date of birth of Naresh Kumar?")
Search([3], "What is the date of birth of Emir Kusturica?")
`</action>`

---

Table 10: Data synthesis prompt template for the Planner module (continued)

## User ##

Observation:
Naresh Kumar was born on 22 December 1928.
Emir Kusturica was born on 24 November 1954.

## Assistant ##

```
<think>
```
Now that I have the birth dates of the two directors. Since Naresh Kumar was born earlier than Emir Kusturica, the director of Ek Paheli was born earlier.
```
</think>
<answer> Ek Paheli </answer>
```

Table 11: Data synthesis prompt template for the Executor module

## System Prompts ##

Your task is to provide reasoning path to answer a question based on some supporting evidence.
Note that there are some disturbance terms within the supporting evidence.
You will be given a question, the answer of the question, and the index of golden supporting evidence to obtain the answer.
You need to:
1. Identify the requested information and summarize the information provided in the golden evidence.
Please put your thoughts between `<think>` and `</think>` tags. Please do NOT consider other supporting evidence other than the golden evidence.
2. Identify the index of the ground truth supporting evidence. Put the index between `<select>` and `</select>` tags.
3. Summarize the answer with one sentence. Put the answer between `<answer>` and `</answer>` tags.

## User ##

Question: Who is the mother of Xawery Żuławski?
Supporting evidence:
[0] (Xawery Żuławski, mother, Małgorzata Braunek)
[1] (Xawery Żuławski, father, Andrzej Żuławski)
[2] (Xawery Żuławski, family, Q63532193)
[3] (Xawery Żuławski, family name, Q56541485)
[4] (Xawery Żuławski, spouse, Maria Strzelecka)
[5] (Xawery Żuławski, date of birth, 1971-12-22T00:00:00Z)
[6] (Xawery Żuławski, sibling, Vincent Zulawski)
[7] (Xawery Żuławski, place of birth, Warsaw)
[8] (Andrzej Żuławski, child, Xawery Żuławski)
[9] (Małgorzata Braunek, child, Xawery Żuławski)
Answer: Małgorzata Braunek
Golden evidence: [0] (Xawery Żuławski, mother, Małgorzata Braunek)

## Assistant ##

```
<think>
```
The question asks me to find the mother of Xawery Żuławski. [0] says Xawery Żuławski's mother Małgorzata Braunek. `</think>`
```
<select> [0] </select>
<answer>
```
The mother of Xawery Żuławski is Małgorzata Braunek. `</answer>`

# H PROMPT TEMPLATES OF RAG AGENT TRAINING

Table 12: Planner Prompt Template for text-based multi-hop question answering

---

**## System Prompts ##**
Your task is to provide a reasoning path to answer a complex question.
You will be given a question and the answer of the question.
You need to:
1. Make a plan about how to answer this question by decompose it into multiple steps.
2. Solve the problem in a step-by-step manner.
Put your thoughts between <think> and </think> tags.
You can use the Search("question") function to get information from supporting evidence.
Please note the the content within the Search function should be a question (interrogative sentence).
When issuing a search query, please refrain from using pronouns like "this", "these", etc., and use the full entity names instead.
Label the final answer within <answer> *** </answer>.

---

**## User ##**
Question: When was the institute that owned The Collegian founded?

---

**## Assistant ##**
<think>
To solve this problem, I need to:
1. Find the institute that owned The Collegian.
2. Figure out when the institute that owns The Collegian was founded.
Now I need to search the information about The Collegian first.
</think>
<action> Search("Which institute owns The Collegian?") </action>

---

**## User ##**
Obs: The Collegian is owned by Houston Baptist University.

---

**## Assistant ##**
<think>
Now that I have the owner of The Collegian, I need to figure out when Houston Baptist University was founded.
</think>
<action> Search("When was Houston Baptist University founded?") </action>

---

**## User ##**
Obs: Houston Baptist University was founded in 1960.

---

**## Assistant ##**
<think>
OK. Now I have all the necessary information to answer the question. The question asking when the institute that owned The Collegian was founded.
</think>
<answer> 1960 </answer>

---

Table 13: Executor Prompt Template for Text-based Multi-hop Question Answering

## System Prompts ##

Your task is to answer a question based on some supporting evidence.

Note that there are some disturbance terms within the supporting evidence.

You will be given a question and a list of supporting evidence.

You need to:

1. Identify the requested information from the question. Review all supporting evidence, summarize the information provided in evidences that support answering the question. Please put your thoughts between `<think>` and `</think>` tags.

2. Identify the index of the golden supporting evidence. Put the index between `<select>` and `</select>` tags. If multiple supporting evidence contain the answer, select all of them. If there are no evidence matches, respond with "No relevant information found." and do not output any other contents.

3. Summarize the answer with one complete declarative sentence. Put the answer between `<answer>` and `</answer>` tags.

Please use the following template:

```
<think> ...  </think>
<select> [X] </select>
<answer> The complete answer sentence.  </answer>
```

## User ##

Question: When was magazine LaIsha founded?

Supporting evidences:

[0] LaIsha: LaÍsha (also known as "For the Woman") is an Israeli magazine for girls and boys.

[1] LaIsha: It has been published on weekly basis since 1947, and is owned by Yedioth Ahronoth media group.

[2] Glamour (magazine): Glamour is a women's magazine published by Condé Nast Publications.

[3] Glamour (magazine): Founded in 1939 and first published in April 1939 in the United States, it was originally called "Glamour of Hollywood"

[4] Salt to the Sea: Salt to the Sea is a 2016 historical fiction young adult novel by Ruta Sepetys.

[5] Salt to the Sea: It tells the story of four individuals in World War II who make their way to the ill-fated MV "Wilhelm Gustloff".

[6] Salt to the Sea: The story also touches on the disappearance of The Amber Room, a work of art stolen by the Nazis that has never been recovered.

[7] MV Wilhelm Gustloff: MV "Wilhelm Gustloff" was a German military transport ship which was sunk on 30 January 1945 by in the Baltic Sea while evacuating German civilians, Nazi officials and military personnel from Gdynia (Gotenhafen) as the Red Army advanced.

[8] MV Wilhelm Gustloff: By one estimate, 9,400 people died, which makes it the largest loss of life in a single ship sinking in history

## Assistant ##

`<think>` The question asks me to determine when the magazine LaIsha was founded. [1] says the magazine LaIsha was founded in 1947. `</think>`

`<select>` [1] `</select>`

`<answer>` LaIsha was founded in 1947. `</answer>`

# I  CASE STUDY

**RQ9: Are RAG agents trained by synthetic reasoning trajectories robust enough when encountering unexpected situations (e.g. incomplete retrieval and failed plan)?** In real-world scenarios, RAG agents may receive negative feedback from the external environment. Specifically, we encounter incomplete retrieval results in 978 out of the 7, 405 questions in the HotpotQA development set. However, our RAG agents is still able to correctly answer 291 of these questions, achieving an accuracy of 29.8%, which demonstrates strong robustness. From examining the actual trajectory outputs, we observe that our RAG agent primarily employs two strategies to cope with incomplete or failed retrieval: (i) rephrasing the sub-question and issuing an additional search query (see Table 14 and 15 for case study), and (ii) making inferences or educated guesses based on its internal knowledge (see Table 15).

Since the datasets (Yang et al., 2018; Trivedi et al., 2022; Ho et al., 2020) do not provide ground-truth intermediate answers, we cannot automatically detect intermediate errors or compute the successful recovery rate. Nevertheless, actual reasoning trajectories obtained from our experiments show that the RAG agent trained by EviPath synthesized data is also able to conduct self-reflection and revise the plan accordingly. An illustrative example is shown in Table 16, where the agent is required to find *the 1995 American black comedy and psychological horror film that Taylor Nichols was part of*. Initially, the agent plans to identify all films that Taylor Nichols has been a part of, and then check if any of the films match the description. However, *none* of the films obtained from the executor satisfies the condition. Then the agent conducts self-reflection and tries to figure out films related to "The Last Days of Disco" (one of the Taylor Nichol's film) but still fails to obtain positive feedback. Hence, the planner of our agent reflects once more and merges the two planning steps into a single step. Finally, the executor returns the desired result, enabling our agent to conclude the correct answer, "*Headless Body in Topless Bar*".

Table 14: Case 1 - Recover from Incomplete Retrieval

---

**## Question ##**
Were Scott Derrickson and Ed Wood of the same nationality?

---

**## Answer:** Yes **##**

---

**## Planner ##**
```
<think>
```
To solve this problem, I need to:
1. Determine the nationality of Scott Derrickson.
2. Determine the nationality of Ed Wood.
3. Compare their nationalities to determine if they were of the same nationality.
I need to search information for both Scott Derrickson and Ed Wood.
```
</think>
<action>
```
Search("What is the nationality of Scott Derrickson?")
Search("What is the nationality of Ed Wood?")
```
</action>
```

---

**## Executor (Retrieval results omitted.) ##**
Obs: No direct information is available about Scott Derrickson's nationality.
Ed Wood is American.

---

**## Planner ##**
```
<think>
```
Since I couldn't find any information about Scott Derrickson's nationality, let me try searching again using different keywords.
```
</think>
<action> Search(``Scott Derrickson country of origin'')
</action>
```

---

**## Executor ##**

---

*Continued on next page...*

Table 14: Case 1 - Recover from Incomplete Retrieval (continued)

Obs: Scott Derrickson is an American director, screenwriter and producer.

## Planner ##
<think>
Now that I know both Scott Derrickson and Ed Wood are Americans, I can conclude that they share the same nationality.
</think>
<answer> Yes </answer>

Table 15: Case 2 - Recover from Incomplete Retrieval

## Question ##
Which television series was part of Cartoon Network's 2017 April Fools' prank and had a song by Justin Roiland played during the third season of the same show?

## Answer:  Rick and Morty  ##

## Planner ##
<think>
To solve this problem, I need to:
1. Identify the television series that was part of Cartoon Network's 2017 April Fools' prank.
2. Confirm that this series had a song by Justin Roiland played during its third season.
Now I need to search the information about Cartoon Network's 2017 April Fools' prank first.
</think>
<action>
Search("Which TV series was part of Cartoon Network's 2017 April Fools' prank?")
</action>

## Executor (Retrieval results omitted.) ##
Obs: No information found about Cartoon Network's 2017 April Fools' prank.

## Planner ##
<think>
Since no direct information is available about Cartoon Network's 2017 April Fools' prank, let me try searching for TV series that were involved in such pranks.
</think>
<action>
Search("TV series involved in Cartoon Network's April Fools' prank")
</action>

## Executor ##
Obs: Mighty Magiswords was likely involved in Cartoon Network's April Fools' prank.

## Planner ##
<think>
Now that I have a potential TV series involved in Cartoon Network's April Fools' prank, I need to confirm that it had a song by Justin Roiland played during its third season.
</think>
<action>
Search("Did Mighty Magiswords have a song by Justin Roiland during its third season?")
</action>

## Executor ##
Obs: There is no information about Mighty Magiswords having a song by Justin Roiland during its third season.

## Planner ##
<think>

Table 15: Case 2 - Recover from Incomplete Retrieval (continued)

However, I know that Rick and Morty has a song by Justin Roiland. Let me see if Rick and Morty was involved in Cartoon Network's April Fools' prank or had a connection to Mighty Magiswords.
```
</think>
<action>
```
Search("Was Rick and Morty involved in Cartoon Network's April Fools' prank?")
```
</action>
```

**## Executor ##**
Obs: Yes, Rick and Morty was involved in Adult Swim's April Fools' prank.

**## Planner ##**
```
<think>
```
Based on the provided information, I can conclude that Rick and Morty is the most likely candidate for the TV series that had a song by Justin Roiland played during its third season, but it was not directly involved in Cartoon Network's April Fools' prank.
```
</think>
<answer> Rick and Morty </answer>
```

Table 16: Case 3 - Recover from the Failed Plan

**## Question ##**
Taylor Nichols was part of the ensemble cast of which 1995 American black comedy and psychological horror film?

**## Answer:** Headless Body in Topless Bar **##**

**## Planner ##**
```
<think>
```
To solve this problem, I need to:
1. Identify the films that Taylor Nichols has been a part of.
2. Check if any of these films match the description of being a 1995 American black comedy and psychological horror film.
Now I need to search the information about Taylor Nichols' filmography.
```
</think>
<action>
```
Search("What films has Taylor Nichols been a part of?")
```
</action>
```

**## Executor (Retrieval results omitted.) ##**
Obs: Taylor Nichols is best known for roles in the Whit Stillman films "Metropolitan", "Barcelona", "The Last Days of Disco", and "Damsels in Distress".

**## Planner ##**
```
<think>
```
Now that I have the list of films associated with Taylor Nichols, I need to check if any of them match the description of being a 1995 American black comedy and psychological horror film.
```
</think>
<action>
```
Search("Is The Last Days of Disco a 1995 American black comedy and psychological horror film?")
```
</action>
```

**## Executor ##**
Obs: The Last Days of Disco is not a 1995 American black comedy and psychological horror film.

**## Planner ##**

Table 16: Case 3 - Recover from the Failed Plan (continued)

```
<think>
```
I need to continue searching other films associated with Taylor Nichols to see if they match the description.
```
</think>
<action>
```
Search("Is Metropolitan a 1995 American black comedy and psychological horror film?")
```
</action>
```

**## Executor ##**
Obs: There is no information indicating that Metropolitan is a 1995 American black comedy and psychological horror film.

**## Planner ##**
```
<think>
```
I need to continue searching other films associated with Taylor Nichols to see if they match the description.
```
</think>
<action>
```
Search("Is Barcelona a 1995 American black comedy and psychological horror film?")
```
</action>
```

**## Executor ##**
Obs: Barcelona is not a 1995 American black comedy and psychological horror film.

**## Planner ##**
```
<think>
```
I need to continue searching other films associated with Taylor Nichols to see if they match the description.
```
</think>
<action>
```
Search("Is Damsels in Distress a 1995 American black comedy and psychological horror film?")
```
</action>
```

**## Executor ##**
Obs: Damsels in Distress is not a 1995 American black comedy and psychological horror film.

**## Planner ##**
```
<think>
```
I have searched all the films associated with Taylor Nichols, but none of them match the description of being a 1995 American black comedy and psychological horror film. However, I noticed that The Last Days of Disco was released in 1998, but it might be related to another film. Let me check again.
```
</think>
<action>
```
Search("Is there any relation between The Last Days of Disco and another film?")
```
</action>
```

**## Executor ##**
Obs: There is no direct relation mentioned between The Last Days of Disco and another film, however, it is loosely based on the writer's travels and experiences in various nightclubs in Manhattan, including Studio 54.

**## Planner ##**
```
<think>
```
Since The Last Days of Disco does not match the description, let me try to find more information about the film that matches the description.
```
</think>
<action>
```
Search("What is the name of the 1995 American black comedy and psychological horror film that Taylor Nichols was part of?")

Table 16: Case 3 - Recover from the Failed Plan (continued)

```
</action>
```

**## Executor ##**
Obs: The film is Headless Body in Topless Bar.

**## Planner ##**
```
<think>
```
OK. Now I have all the necessary information to answer the question. The question asking what is the 1995 American black comedy and psychological horror film that Taylor Nichols was part of.
```
</think>
<answer> Headless Body in Topless Bar </answer>
```

## THE USE OF LARGE LANGUAGE MODELS

We use LLMs, namely GPT-4o, GPT-5 and Gemini 2.5 Pro, as general-purpose assist tools for grammar checking and paper polishing. We also use GPT-4o to re-examine some of the baseline methods.

