# OpenReview forum: "From Evidence to Trajectory: Abductive Reasoning Path Synthesis for Training Retrieval-Augmented Generation Agents"
_ICLR.cc/2026/Conference — Submitted to ICLR 2026_

### Official Review · Reviewer_jQ3j · 2025-10-18

**Soundness:** 3
**Presentation:** 3
**Contribution:** 2
**Rating:** 2
**Confidence:** 4

**Summary:**

This paper proposes EviPath, an evidence-anchored reasoning path synthesis paradigm for RAG agent, which comprises three stages: 1) Abductive Subtask Planning: decomposes the problem into sub-questions and iteratively plans an optimal solution path based on the dependencies between them; 2) Faithful Sub-question Answering: uses supporting evidence to construct a proxy environment to generate reasoning thoughts and answers for each sub-question, and 3) Conversational Fine-Tuning: formats the complete agent-environment interaction trajectory into a dialogue format suitable for SFT. Experiments show the effectiveness of various models trained on data synthesized through EviPath. The paper also answers six research questions based on many ablation studies.

**Strengths:**

1. The paper tries to transform the traditional forward trajectory synthesis paradigm to an abductive reverse process.
2. The introduction of intermediate answer synthesis can effectively boost the quality of synthetic trajectories.
3. The figures shown in this paper are easy to understand and can clearly present the pipeline of the proposed method.
4. The main experiments and ablation studies are comprehensive.

**Weaknesses:**

1. My main concern lies in the scalability of the methods in this paper. As EviPath heavily relies on the presence of a large number of questions, golden evidence, and answers, these elements can be found in common multi-hop QA datasets. However, in real-world scenarios, especially in specific domains, it is often challenging to obtain a substantial amount of high-quality answers and evidence, and sometimes there may not be enough questions available. In such cases, EviPath may prove to be inadequate.

2. The intermediate processes in this paper are all synthesized in a simulated environment based on gold evidence. The entire process assumes that the synthesized intermediate answers and thoughts are correct, which introduces several issues: 1) Can gold evidence truly simulate a real retrieval environment? 2) Are the synthesized intermediate thoughts and answers genuinely accurate? 3) Models trained in this manner have only encountered positive environmental feedback. Once they generate errors during inference or receive incorrect external signals, the model is prone to failure.

3. There are doubts about the fairness of the baseline comparisons: 1) The paper introduces the signal of gold evidence. Do all baselines adhere to the same setup? 2) The training in this paper is conducted on 265K data points. Have all baselines been trained on the same amount of data? 3) How would methods like HippoRAG, IRCoT, EfficientRAG perform on models such as GPT-4o/DeepSeek-V3/DeepSeek-R1, given that prompt-based baseline models often have weaker backbones?

4. The datasets tested in this paper only involve retrieval and do not seem to include web search and knowledge graph as shown in Figure 2.

5. The datasets used to test the generalization in RQ3 still exhibit strong similarities to the training set, which may not be sufficient to draw conclusions.

**Questions:**

1. What are the reproduction details of RAG Agents and concurrent works in Table 1, including the composition and scale of the training data, as well as the information contained in the prompts?

2. Is there an analysis regarding the robustness of the trained models, such as their behavior when encountering situations not seen during training (e.g., generating incorrect intermediate answers, encountering retrieval errors)?

3. Given that LLMs today already possess strong foundational abilities in multi-hop QA, and RL algorithms can enable models to self-explore trajectories relying solely on the final reward and engage in self-reflection and correction, is it necessary to synthesize a large amount of trajectory data in this scenario?

---

> ### Author Response · Authors · 2025-11-19
> **Response to Reviewer jQ3j for Weakness 1 and 2**
>
> Thank you for your acknowledgement of the strengths and contributions of our manuscript.
>
> ### Reply to Weakness 1:
> In order to address the concern on the over-reliance of golden evidence provided in relevant datasets, we have conducted ablational experiments and reported the results in our ***Table 1*** of our original/revised manuscript (Refer to the results leaded by “-w/o supporting facts). Under this setting, the EM score drops by 4.1% compared to using evidence-augmented synthetic data, but ***still surpasses the best baseline by 10.6%***.
>
> Given the double-digit performance improvements, we can conclude that EviPath still works well in the absence of golden evidence.
>
> ### Reply to Weakness 2:
>
> ***[2.1]*** The simulated retrieval environment includes both golden evidence and distractive passages, which is similar to real retrieval results where some pieces of evidence are supportive and others are not. We acknowledge that, in real-world scenarios, retrieval may sometimes fail to return any truly supportive evidence. However, *without access to relevant contextual information, an LLM cannot be expected to reliably answer open-domain factual QA questions*.
>
> ***[2.2]*** Similar to the RL-based baselines, we cannot guarantee that trajectories with correct final answers always contain fully correct intermediate steps. Moreover, since the datasets do not provide annotations for intermediate answers, we are unable to directly evaluate this aspect. Nevertheless, the substantial performance gains achieved with a limited amount of training data (Figure 5) indirectly suggest that the quality of the synthetic data generated by EviPath is reasonably high.
>
> ***[2.3]*** We agree that models trained with SFT and our synthetic data only observe positive feedback during training. However, this does not prevent our RAG agent from being robust to incomplete retrieval. Based on a manual inspection of the outputs on the HotpotQA development set, we identified incomplete retrieval in 978 out of 7,405 questions, yet our RAG agent still answered 291 of these correctly. We have conducted a case study regarding this issue, which is included in Appendix H of our paper.

---

> ### Author Response · Authors · 2025-11-19
> **Response to Reviewer jQ3j for Weakness 3**
>
> ### Reply to Weakness 3:
>
> ***[3.1]*** Existing baselines, especially outcome-rewarded RL methods, do not natively support planning that is explicitly guided by gold evidence. Incorporating both answers and evidences into the trajectory rollout process of these RL methods would require substantial changes to their algorithms, workflows and implementations. In our view, this would already constitute a new line of work, which goes beyond the original design, scope, and capacity of the existing baselines.
>
> The effective use of gold evidence to reverse-construct concise, interactive reasoning trajectories for offline training is one of the main contributions, and key advantages of our approach. *It is not a minor setting choice that can be trivially plugged into existing RL baselines, but rather a central design of EviPath itself.
>
> We have also conducted ablation experiments to assess the effect of introducing gold evidence. As detailed in our response to Weakness 1, even without gold evidence, EviPath still achieves a 10.6% absolute EM gain, showing that the success of EviPath cannot be fully attributed to the usage of extra information.
>
> ***[3.2]*** We thank the reviewer for raising this concern about training data scale.
>
> (1) ***Clarification on the 265K data points***: We would first like to clarify that we did ***not*** use all 265k trajectories to train a single QA agent. Trajectories from different datasets are never mixed. The actual number of training samples adopted for the HotpotQA, MuSiQue, and 2wiki datasets are 86k, 20k, and 159k, respectively.
>
> (2) ***Baseline settings follow their original papers***: The reported results for baseline methods are taken directly from their original papers, as stated in the caption of Table 1. We do not re-train these baselines with our data; instead, we follow their published settings. The detailed training configurations and prompt usage for each baseline can be found in the corresponding original works.
>
> (3) ***Ablation on data scale - EviPath works well with much less data***: To address potential concerns about the scale of our training data, we conducted an ablation on training steps. We extracted checkpoints between 200 and 2,000 training steps (step size 200) and evaluated them (see Figure 5 and RQ6 in the paper). Training for ***2,000 steps (≈20,000 questions)*** achieves performance within ***about 1%*** of using the full dataset. This scale (≈20k questions) is roughly comparable to the data size used by Graph-R1 (30,720 examples) and is in the same order as other baselines.
>
> (4) ***Using same number of question is not a fair notion of cost for RL baselines***
>
> ***RL algorithms like GRPO typically sample and train on multiple trajectories per question*** (e.g., 5 trajectories), which makes their ***effective training budget*** much larger than the raw number of questions suggests.
>
> For example, ***Graph-R1*** is trained on 30,720 examples for 3 epochs and samples 5 trajectories per question, resulting in roughly 460k trajectories used for training. This effective training cost is ***significantly higher*** than that of EviPath, even when EviPath uses its full dataset. Therefore, EviPath’s advantages cannot be attributed simply to using more data; in fact, RL baselines may consume substantially more trajectories during training.
>
> ***[3.3]*** Regarding the selection of backbone models for prompt-based baselines.
>
> **(1) We follow the original backbone choices of HippoRAG, IRCoT, and EfficientRAG**
>
> For HippoRAG, IRCoT, and EfficientRAG, we directly report the results from their original papers and keep their default backbone LLMs unchanged:
>
> - **HippoRAG**: GPT‑3
> - **IRCoT**: GPT‑3.5
> - **EfficientRAG**: fine‑tuned Llama‑3‑8B
>
> We do ***not*** modify their LLM choices, and we do ***not*** re‑run or downscale these methods with weaker models. Thus, their reported performance is exactly as claimed by the original authors.
>
> **(2) Their backbones are not weaker than ours; in many cases they are stronger**
>
> The backbone LLMs used in these baselines (GPT‑3, GPT‑3.5, and Llama‑3‑8B) are comparable to, or in practice often stronger than, the models we use in EviPath (Qwen2.5‑7B, Llama 3.2‑1B/3B/ 3.1-8B). Therefore, using the original results of HippoRAG, IRCoT, and EfficientRAG does not rely on giving them “weaker backbones”; if anything, these baselines benefit from relatively strong or even stronger LLMs.
>
> Consequently, the comparison does not hinge on handicapping prompt-based methods via weaker backbones, and the soundness of our main claims is not undermined by the backbone choices for these baselines.

---

> ### Author Response · Authors · 2025-11-20
> **Response to Reviewer jQ3j for Weakness 4-5, Question 1-3**
>
> ### Reply to Weakness 4:
>
> We would like to clarify that our experiments **do include knowledge graph interaction**.
> - For the **2WikiMQA** and **QALD‑10** datasets, our agent **interacts with the external Wikidata Knowledge Graph** to retrieve relevant triples.
> - This interaction is implemented via the **official Web API provided by Wikidata** (see Appendix E of the paper for our original description).
> - The detailed interaction patterns and query formats for Wikidata KG are documented in **Table 9 in Appendix F**.
>
> During evaluation, **neither the answer nor the golden triples is revealed** to the agent. The agent only has access to the retrieval and KG query mechanisms described above, consistent with the intended setting in Figure 2.
>
> ### Response to Weakness 5:
>
> For our OOD generalization study in RQ3, we follow the **same evaluation convention** adopted by recent baselines such as **Graph‑R1** and **Mujica‑MyGO**, where models are typically trained on **one open‑domain QA dataset** and then tested on **another open‑domain QA dataset**.
>
> To the best of our knowledge, existing **outcome‑rewarded RL–based agentic RAG baselines** trained on *general QA* **do not** report cross‑domain generalization results to **math** or **coding** tasks. Thus, our setup is therefore consistent with the prevailing practice in this line of work.
>
> ### Response to Question 1:
>
> Please refer to our reply to Weakness 3.2.
>
> ### Response to Question 2:
>
> Thank you for this constructive suggestion.
>
> Regarding the robustness along the axis of retrieval errors,  we checked the trajectories outputted from the evaluation experiments. Specifically, on the **HotpotQA development set**, we observed **incomplete retrieval**  in **978 out of 7,405** questions, yet our RAG agent still **answered 291 of these correctly**, achieving an accuracy of 29.8%.
>
> From a manual inspection of the trajectories, we found that our agent typically adopts two main strategies to cope with incomplete or failed retrieval:
> 1. **Rephrasing and re‑retrieval:** reformulating the sub‑question and issuing another retrieval request.
> 2. **Inference / educated guessing:** leveraging its internal knowledge to infer plausible answers when evidence is missing.
>
> Regarding robustness to **incorrect intermediate answers**, the datasets we use do **not** provide ground‑truth annotations for intermediate reasoning steps. Consequently, we are unable to automatically detect all intermediate errors and compute a precise “recovery rate”. However, we provide a qualitative **case study** to illustrate such behaviors.
>
> For more detailed analyses related to robustness and error cases, please refer to **Table 13-15, Appendix H** of our revised manuscript.
>
> ### Response to Question 3:
> We believe that data curation and the synthesis of high‑quality reasoning trajectories remain important, even in the era of RL‑based methods.
>
> **(1) Current LLMs are still far from solving complex problems reliably**
>
> While modern LLMs have much stronger foundational abilities than earlier models, their multi‑hop QA performance is still not sufficiently desirable. For example, we evaluated the pre‑trained **Llama‑3.1‑70B** model on multi‑hop QA (see Table 1), and its performance on **MuSiQue** remains very low (**EM = 13.1**), suggesting that the agentic multi‑hop reasoning capability is still limited.
>
> **(2) SFT on synthetic trajectories can be more efficient than outcome‑rewarded RL**
> - RL methods such as **GRPO,GSPO,DAPO, etc...** typically require **repeated rollouts** and **multiple trajectories per question** to obtain accurate advantage estimates, which makes training costly.
> - In contrast, SFT on EviPath’s synthetic trajectories **eliminates the need for repeated online sampling**, and directly provides supervised signals for intermediate steps.
>
> **(3) RL is limited by the LLM’s inherent capability ceiling**
> Specifically, if an LLM is **intrinsically unable** to solve a certain class of problems, it will struggle to obtain **meaningful positive rewards** through self‑exploration alone.  In such regimes, RL may be ineffective because the LLM rarely, if ever, discovers successful trajectories from which to learn. In these cases, **guidance from gold evidence and gold trajectories** is crucial: it can bootstrap the model into regimes where RL becomes viable, or even provide capabilities that RL alone would not easily acquire.
>
> In summary, despite the progress of strong LLM backbones and RL algorithms, **high‑quality synthetic reasoning trajectories remain necessary and complementary**: they improve sample efficiency, reduce reliance on expensive rollouts, and help overcome the inherent limitations of purely reward‑driven self‑exploration.
>
> We hope our rebuttal can address your doubts and concerns. We are looking forward to receiving your feedback.

---

> > ### Comment · Reviewer_jQ3j · 2025-11-25
> >
> > Thank you for the detailed feedback. I still have the following concerns that prevent me from considering this paper as acceptable:
> >
> > 1. I believe that the foundation for synthesizing a large amount of data in this paper relies on leveraging existing question and answer pairs. However, obtaining high-quality QA pairs in real-world scenarios is challenging, which raises doubts about the scalability of the method proposed in this paper. It seems that the authors have only responded to this issue concerning gold evidence.
> >
> > 2. The accuracy of 29.8% on the incomplete retrieval trajectory is significantly lower than the overall standard, casting doubt on the robustness of the approach.
> >
> > 3. Unfair comparisons have been made. While weaker baselines like CoT and RAG are backed by GPT-4o, stronger baselines like RAFT, HippoRAG, and IRCoT are supported by models such as Llama-8B, GPT-3.5, and GPT-3. This discrepancy raises suspicions of unfair comparisons. For instance, IRCoT's performance on HotPotQA with GPT-3 is already close to EviPath's, and using GPT-4o may not necessarily lead EviPath to achieve the state-of-the-art result.
> >
> > 4. Being consistent with the prevailing practice in this field does not necessarily imply that the setup is rational. The lack of convincing evidence regarding whether the system truly has out-of-distribution generalization ability remains a concern.

---

> > > ### Author Response · Authors · 2025-12-02
> > > **Response to Concerns 1-2**
> > >
> > > We would like to thank the reviewer for his/er response. We are also glad to provide further clarification regarding the additional concerns.
> > >
> > > Before addressing the detailed issues, we would like to note that EviPath is designed to ***synthesize reasoning trajectories*** for ***training*** small-scale LLMs, so that they acquire strong agentic reasoning and question-answering capabilities.
> > >
> > > ### ***1. Reliance on Question-Answering Pairs.***
> > >
> > > We would like to clarify three points regarding the role of QA pairs.
> > >
> > > ### ***(a) For weak small-scale LLMs, supervision from QA pairs is essential, not optional.***
> > >
> > > ***Our target models*** are relatively ***weak LLMs*** that *do not possess adequate question-answering capabilities by default*. For such models, in-context demonstrations and RAG pipelines are insufficient to enable accurate question answering. For instance, an pre-trained Llama3.1-8B-Instruct model can only achieve EM=19.5 on HotpotQA dataset with few-shot demo.
> > > In this case, using QA pairs becomes a necessary foundation. EviPath builds on this minimal requirement and focuses on constructing dense, interactive reasoning supervision.
> > >
> > > The authors are welcome to compare any training methods that are able to ***endow weak (1B,3B,8B) models*** with adequate QA capabilities ***without the help of QA pairs***.
> > >
> > > ### ***(b) EviPath does NOT require large-scale, perfectly curated QA pairs.***
> > > In order to show the low reliance of the large number of QA pairs, we *re-conducted experiments* by using ***10000*** questions only, which is about ***1/17 of the size of 2WikiMQA dataset***. With such a limited number of questions, EviPath can still outperform the best baseline with ***9.0% absolute EM improvement***. Moreover, the model performance *scales up* as the amount of training data increases (see Figure 5 in our Paper), which demonstrates the effectiveness of our method.
> > >
> > > |                                          | HotpotQA |      | MuSiQue |      | 2WikiMQA |      | Average |      |
> > > |------------------------------------------|:--------:|:----:|:-------:|:----:|:--------:|:----:|:-------:|:----:|
> > > |                                          |    EM    |  F1  |    EM   |  F1  |    EM    |  F1  |    EM   |  F1  |
> > > | EviPath w/ 10000 QA Pairs, no Golden evid.       |   50.9   | 64.3 |   31.7  | 42.4 |   89.6   | 92.6 |   57.4  | 66.4 |
> > > | EviPath All QA Pairs, noGolden evid. |   51.8   | 65.0 |   34.0  | 44.3 |   91.1   | 93.4 |   59.0  | 67.6 |
> > > | EviPath (full)                           |   53.8   | 66.4 |   44.3  | 54.6 |   91.3   | 93.6 |   63.1  | 71.5 |
> > >
> > > ### ***(c) Obtaining high-quality QA pairs is not expensive.***
> > > Question answering datasets ***are not necessarily to be manually curated*** by human experts. For instance, the high-quality 2WikiMultihopQA datasets, with ***167k QA pairs***, are constructed ***in an automatic manner***. Humans are only responsible for designing the initial templates and logical rules.
> > > ***In the LLM era***, constructing question-answer pairs will be ***much more easier***.
> > >
> > > ---
> > >
> > > ### ***2. Regarding the comment ```The accuracy of 29.8% on the incomplete retrieval trajectory is significantly lower than the overall standard, casting doubt on the robustness of the approach.```***
> > >
> > > ***(a)*** When the retriever or external environment *fails to return the necessary supporting context*, ***it is unrealistic to expect the LLM to be able to answer the question correctly***. In case, the model can only ***rely on knowledge memorized during pre-training***, which is inherently limited for small-scale LLMs. Since this scenario primarily assesses the model’s memorization capacity, it would be ***inappropriate*** to conclude that our proposed synthetic data methodology is unreliable.
> > >
> > > ***(b) Given that the overall EM on HotpotQA is only 53.8, achieving 29.8 EM under incomplete retrieval with an 8B LLM–driven RAG agent is already a strong result.*** For comparison, ***even GPT 4o***, which has substantially stronger internal memory, ***achieves only 29.4 EM*** in a chain-of-thought setting *without supporting evidence*. The good performance indicates that EviPath-trained agent remains competitive and reasonably robust, even under deliberately handicapped retrieval conditions.

---

> ### Author Response · Authors · 2025-12-02
> **Response to Concerns 3-4**
>
> ### ***3. Regarding the comment ``"Unfair comparisons have been made"``***
>
> ### ***(i) Regarding the nature of baselines that are asked for comparison***
>
> First, we would like to clarify that EviPath proposes ***a data synthesis framework*** for RAG agent development, which is able to ***train 1B / 3B / 7-8B small scale LLMs*** with strong QA capability. In case, EviPath necessarily requires the backbone LLM the RAG agent to be “***open-sourced***” and “***trainable***”.
>
> In contrast, ***pipeline RAG baselines (e.g. HippoRAG, IRCoT)*** the reviewer requested to compare rely on large-scale commercial LLMs that are “***close-sourced***” and “***untrainable***”. Hence, ***these method are not able toimprove the intrinsic reasoning/QA capability of weak backbone LLMs***. Therefore, these baselines are provided ***for reference only***.
>
> Besides, the ***RAFT*** baseline the reviewer have mentioned is a “***LLM training***” method, which ***cannot be adapted to GPT-4o*** as requested. The original backbone of this baseline is ```Llama3-8B```, which is ***similar to / stronger than*** the ```Llama3.1-8B / Llama3.2-3B``` model we adopted. Given the superior performance gain, we can conclude that EviPath significantly outperforms RAFT.
> Therefore, it would be ***inappropriate*** to conclude that `` Unfair comparisons have been made``.
>
> ### ***(ii) Stronger backbones do not allow SOTA pipeline RAG systems to surpass the EviPath-trained 8B agent***
>
> We understand that using more advanced backbone LLM may improve the performance of pipeline RAG systems. Therefore, we conducted additional experiments for ***two highly-competitive pipeline RAG method HippoRAG and IRCoT*** with ***GPT-4o*** (see Table below). We have included the results and related analysis in our revised manuscript ***(Appendix F)***.
>
> |                               | HotpotQA |      | MuSiQue |      | 2WikiMQA |      | Average |      |
> |-------------------------------|:--------:|:----:|:-------:|:----:|:--------:|:----:|:-------:|:----:|
> |                               |    EM    |  F1  |    EM   |  F1  |    EM    |  F1  |    EM   |  F1  |
> | IRCoT (GPT3)                  |   49.3   | 60.7 |   26.5  | 36.5 |   57.7   | 68.0 |   44.5  | 55.1 |
> | IRCoT (GPT-4o)                |   49.0   | 64.3 |   31.2  | 45.4 |   58.0   | 71.9 |   46.1  | 60.5 |
> | HippoRAG (GPT3.5)             |   41.8   | 55.0 |   19.2  | 29.8 |   46.6   | 59.5 |   35.9  | 48.1 |
> | HippoRAG (GPT-4o)             |   43.7   | 57.7 |   26.9  | 39.0 |   59.4   | 67.9 |   43.3  | 54.9 |
> | IRCoT+HippoRAG (GPT3.5)       |   45.7   | 59.2 |   21.9  | 33.3 |   47.7   | 62.7 |   38.4  | 51.7 |
> | IRCoT+HippoRAG (GPT-4o)       |   49.2   | 64.0 |   31.5  | 44.2 |   68.0   | 77.2 |   49.6  | 61.8 |
> | EviPath (w/o Golden Evidence) |   51.8   | 65.0 |   34.0  | 44.3 |   91.1   | 93.4 |   59.0  | 67.6 |
> | EviPath                       |   53.8   | 66.4 |   44.3  | 54.6 |   91.3   | 93.6 |   63.1  | 71.5 |
>
> The results are clear:
>
> - Replacing GPT‑3 / GPT‑3.5 with GPT‑4o **does** bring performance gains to HippoRAG and IRCoT.
> - However, **even with GPT‑4o** and even after combining IRCoT’s chain-based retrieval with HippoRAG’s memory indexing, the best pipeline RAG configuration still **lags behind** the EviPath‑trained 8B RAG agent by **13.5 EM** on avg.
> - Notably, **EviPath (w/o Golden Evidence)** already outperforms all GPT‑4o‑based pipeline RAG baselines, further underscoring the strength of our data synthesis and training framework.
>
> (*Note: We have replicated the experiments and confirmed that when IRCoT w/ GPT-4o will **not** surpass IRCoT w/ GPT-3 in terms of EM.*)
>
> > “For instance, IRCoT’s performance on HotPotQA with GPT‑3 is already close to EviPath’s, and using GPT‑4o may not necessarily lead EviPath to achieve the state-of-the-art result.”
>
> Empirically, when using GPT‑4o as requested, ***the reviewer’s concern shown above*** ***does not align with the actual experimental findings***.
>
> ### ***4. Regarding the comment ```"Being consistent with the prevailing practice in this field does not necessarily imply that the setup is rational. The lack of convincing evidence regarding whether the system truly has out-of-distribution generalization ability remains a concern."```***
>
> We would like to clarify that, by *following the prevailing evaluation practices in this field*, our setup does allow ***a fair and meaningful comparison***, and clearly demonstrates the advantages of our method over strong baselines under commonly accepted conditions.
>
> We also note that “OOD generalization” is not uniquely defined in this area. Therefore, we ***welcome specific, concrete suggestions*** for OOD evaluation protocols or benchmarks that the reviewer considers more appropriate (e.g., particular datasets, distribution shifts, or task variants), and we are open to test or discuss it.

---

### Official Review · Reviewer_SxGi · 2025-10-31

**Soundness:** 2
**Presentation:** 3
**Contribution:** 2
**Rating:** 6
**Confidence:** 3

**Summary:**

The paper addresses the lack of step-wise supervision in current retrieval-augmented generation (RAG) agents and proposes EviPath, an evidence, anchored reasoning path synthesis framework grounded in abductive reasoning. By applying abductive reasoning, **EviPath** reversely constructs multi-step reasoning trajectories from existing QA pairs and supporting evidence, encompassing three stages: subtask planning, sub-question answering, and conversational fine-tuning. Experimental results demonstrate that models trained on EviPath-synthesized data achieve significant improvements on multi-hop QA benchmarks, validating the effectiveness of data-driven process supervision in enhancing the reasoning capabilities of RAG agents.

**Strengths:**

1. The paper innovatively introduces abductive reasoning into the process of reasoning trajectory generation, providing a complementary paradigm for RAG training that differs from traditional CoT and RL methods.
2. It constructs a dataset containing 265k high-quality reasoning trajectories and provides two types of structured prompt templates, planner and executor, offering effective support for process-level supervision.
3. The paper conducts a systematic comparison with 24 baseline methods in the experiments.

**Weaknesses:**

**1. The theoretical depth and verifiability of the contribution on abductive reasoning are insufficient.**

a. The paper claims that “We are the first to formulate the synthesis of reasoning paths for RAG agents as an abductive reasoning problem” and describes the process of constructing reasoning trajectories backward from the final answer as “abductive subtask planning.” However, in terms of methodological implementation, the actual process still relies on a large language model to generate sub-question decompositions based on the final answer and the evidence set, and subsequently produce the thought and action fields.

b. The paper mentions that golden evidence for sub-questions is selected based on a cosine similarity threshold of 0.9, but it does not analyze the rationale or sensitivity of this threshold. The absence of threshold ablation or robustness experiments makes this step appear arbitrary, and its impact on factual precision and recall remains unclear. Although the paper refers to an “optimal, dependency-aware reasoning plan” and “minimally sufficient w.r.t. F̂,” it does not provide objective criteria for determining what constitutes “optimality” or “minimal sufficiency,” nor does it explain how the search space is pruned or how the process recovers from failures. Consequently, the so-called abductive reasoning in this paper resembles an LLM-driven controlled trajectory reconstruction process rather than a formally constrained abductive reasoning algorithm with verifiable optimality.

**2. Strong reliance on the“golden evidence subset F̂”**

Both key stages of EviPath assume access to a golden evidence subset F̂ that is highly relevant to the question. In Stage 1 (Abductive Subtask Planning), the model directly uses the final answer and F̂ to reverse-generate sub-questions. In Stage 2 (Faithful Sub-question Answering), the framework bypasses real retrieval by constructing a simulated environment, treating the complete set of supporting facts F as the retrieval corpus and selecting, for each sub-question, the subset F̂ᵢ most similar to its intermediate answer.

However, in realistic open-domain RAG settings, retrievers often make errors and rarely have access to true golden facts. The authors construct this RAG agent training paradigm entirely based on simulated environments and synthetic data, yet the experiments do not verify its robustness or transferability in real-world open-domain scenarios, nor do they provide any evaluation of the constructed environment's quality.

**3. Ablation studies are insufficient in depth and coverage**

The paper does not present behavioral case studies comparing planner-only and executor-only settings. For example, it remains unclear whether there are interpretable failure cases such as “retrieving the correct evidence but the planner following an incorrect reasoning path” or “the planner generating a correct plan but the executor producing a wrong answer.” The paper also does not explicitly explain why the planner is considered the main bottleneck compared to the executor. Overall, it lacks qualitative analysis and interpretable examples to substantiate these claims.

**4. Lack of reproducibility and resource availability**

The paper does not provide code, and critical implementation details in the main text and appendix (such as generation parameters including temperature, top-p, max tokens, and n) are insufficiently described, making it difficult for reviewers to independently reproduce the results.

The dataset section only reports the overall scale (265k trajectories) but does not present specific cases or any quality evaluation metrics to demonstrate the validity and reasonableness of the synthesized trajectories.

**Questions:**

**1. Comparison with RL-based methods**

The paper repeatedly claims that the proposed EviPath + SFT approach significantly outperforms RL-based RAG agents such as GRPO, Search-R1, and Graph-R1. However, it does not clearly specify whether these RL baselines were evaluated under the same model scale, training budget, and data size. Given the reported performance gap of up to +14.7 EM, the authors should further clarify the fairness of these comparisons.

**2. On the difference in SFT data density**

Could the performance improvement of EviPath primarily stem from denser supervision (265k complete reasoning trajectories) rather than the method itself? The authors are encouraged to conduct controlled experiments, such as reducing the number of trajectories or removing parts of intermediate annotations, to verify the true source of the observed performance gains.

**3. On the definition and substance of “agentic RAG”**

The paper repeatedly emphasizes that EviPath trains RAG agents with agentic reasoning capabilities, yet the overall pipeline is entirely offline, involving only trajectory synthesis without real environmental feedback, state updates, or tool interaction. The authors should clearly define what constitutes agentic reasoning in the context of this paper and clarify how EviPath fundamentally differs from a workflow-based RAG system.

---

> ### Author Response · Authors · 2025-11-19
> **Reply to Reviewer SxGi for Weakness 1-3**
>
> Thank you for your acknowledgement of the strength and contribution of our manuscript.
>
> ### Reply to Weakness 1:
> 1a. We would like to clarify that term “abductive reasoning” refers to *the reasoning process of providing explanations or hypotheses for observations* [1] [2], which "providing rationales and hypothetical retrieval actions" lie in the scope. ***Abductive reasoning*** can be formulated with rigorous logical reasoning algorithms, but it ***does not naturally limit the specific methods of implementation***. For example, the process can be modelled by frontier neural networks like language models [3] and diffusion models [2].
>
> 1b. We adopt a high cosine similarity threshold of 0.9 since most sub-questions only require a single piece of evidence to be answered. However, there exist very few cases that the LLM need to combine knowledge from multiple sources to answer a single sub-question. A good example could be asking for the nationality of a person who has dual nationalities. Given the excellent preliminary results, we believe 0.9 is an appropriate value for threshold $\tau$.
>
> We agree that the claim regarding “optimality” may cause ambiguity. *Since existing multi-hop question answering datasets ***did not*** provide any ground-truth plans or intermediate answers for sub-questions, it is ***not possible*** to rigorously assess optimality.*
>
> In addition, we would like to clarify that ***the ground truth evidence set*** $\hat{\mathcal{F}}$ provided in dataset ***is minimally sufficient to answer the question***. Therefore, it is expected that *conducting abductive reasoning on* $\hat{\mathcal{F}}$ is able to ***yield concise reasoning trajectories with few reasoning steps***. We have made corresponding adjustments to our statements in Section 3.2 of our revised manuscript.
>
> ### Reply to Weakness 2:
> First, we shall emphasize that ***during evaluation***, we adopt an “***open-domain***” setting. Therefore, ***our RAG agent must interact with real environment***, which ***will suffer from negative feedback***. Hence, the significant performance improvement proves the ***robustness*** and nice ***transferability*** in real-world scenarios.
>
> In addition, we have also conducted the data synthesis process by ***removing all supporting evidence***, and ***reported the results in Table 1*** of our original manuscript as "w/o supporting facts''. Under this setting, our agent ***still surpasses the best baseline by 10.6%***. Therefore, we can conclude that our proposed method does ***not*** strongly rely on the existence of ground truth annotations.
>
> The construction of simulated environments is only applicable to the training trajectory synthesis phase. For each training question, we leverage the supporting facts $\mathcal{F}$ set as the simulated environment, which the ground truth evidence $\hat{\mathcal{F}}$ is guaranteed to be included. Therefore, we can consider the simulated environment sufficiently reliable for our purposes.
>
> ### Reply to Weakness 3:
> Thank you for your suggestion, we have conducted a ***case study*** to illustrate ***how our RAG agent suffers and recovers from incomplete retrieval results and/or inappropriate plans***. For details, we suggest you to refer to the ***Appendix H*** in our revised manuscript.
>
> Regarding the rationale of concluding the planner is the main bottleneck compared to the executor, we have conducted ***ablation studies*** and ***reported the results in the Table 1*** of our original manuscript. Detailed analysis can be found in RQ2, Section 5.4.
>
> Here, we would like to highlight some of the observations here: ***If we replace the planner with a pre-trained LLM, the performance degradation will be significantly more severe than replacing the executor module***. This empirically demonstrates that the planning capability is the main bottleneck for a pre-trained LLM to be leveraged in developing Agents.
> |     F1 scores                   |     HotpotQA    |     MuSiQue    |     2WikiMQA    |     Avg.    |
> |---------------------------------|:---------------:|:--------------:|:---------------:|:-----------:|
> |     w/o planner fine-tuning     |       54.3      |       37.7     |       50.4      |     47.5    |
> |     w/o executor fine-tuning    |       61.8      |       41.9     |       91.6      |     65.1    |
> |     EviPath                     |       66.4      |       54.6     |       93.6      |     71.5    |
>
> References:
>
> [1] Bai et al., Advancing Abductive Reasoning in Knowledge Graphs through Complex Logical Hypothesis Generation (ACL2024)
>
> [2] Bai et al., Unifying Deductive and Abductive Reasoning in Knowledge Graphs with Masked Diffusion Model
>
> [3] Du et al., Learning Event Graph Knowledge for Abductive Reasoning (ACL 2021)

---

> ### Author Response · Authors · 2025-11-19
> **Reply to Reviewer SxGi for Weakness 4 and Question 1-3**
>
> ### Reply to Weakness 4:
>
> We agree that reproducibility is important and would like to provide the following hyperparameters for reference: ***top_p = 0.8, temperature = 0.7, max_tokens = 4096, and repetition_penalty = 1.05***. We utilize the same set of hyper-parameters for reasoning trajectory synthesis and RAG agent evaluation. These additional hyper-parameters settings are included in ***Table 7, Appendix E***.
>
> Due to institutional internal policy and concerns about premature misuse, we are not able to release our code and data at this stage. However, ***we plan to make both the code and data publicly available upon acceptance of this manuscript***.
>
> ### Reply to Question 1:
> We would like to clarify that all results of baseline methods are referred from  ***their original papers*** (see the ***caption*** of ***Table 1*** for our original claim).  We reported the backbone LLMs each baseline method has adopted, and would like to suggest readers to refer to their papers for detailed experimental settings.
>
> In order to investigate the impact of training data size on the experimental results, we extracted all checkpoints between 200 and 2000 training steps at intervals of 200 steps and evaluated them. ***Figure 5*** of our original/revised manuscript shows the experimental results. Indeed, ***training for 2,000 steps (20,000 questions) achieves performance that is within about 1% of the performance achieved using the full dataset*** (see RQ6 for our original statements). This scale is roughly comparable to the data size used in Graph-R1 (30,720 examples) and other baselines.
>
> We would like to mention that directly comparing EviPath with GRPO-based agents and other RL-based methods *under the same number of questions (examples)* can be somewhat *misleading* in terms of ***effective training/exploration cost***.  RL algorithms like GRPO typically sample and train on multiple trajectories (e.g., 5) for each question, which results in a much larger number of trajectories being used during training than the raw number of questions might suggest.
> *For example, ***Graph-R1*** is trained on 30,720 examples from 6 datasets for 3 epochs and samples 5 trajectories per question, resulting in roughly ***460k trajectories*** used for actual training.* In contrast, ***we synthesize only 86k, 20k, and 159k trajectories*** for the HotpotQA, MuSiQue, and 2Wiki datasets, respectively. Each trajectory is trained *only twice*. In order to prevent potential cross-dataset data leakage, trajectories from different datasets are never mixed. Therefore, ***the advantage of EviPath cannot be simply attributed to a larger amount of data or higher training cost***.
>
> ### Reply to Question 2:
>
> Thank you for your constructive suggestions. First, we would like to clarify that we ***did not*** use all 265k trajectories to train our QA agent. Trajectories from different datasets are ***never*** mixed together.
>
> In addition, *we have already conducted the controlled experiments that you have mentioned and reported the results in our original manuscript*.
>
> Specifically, we ***reduced the number of trajectories used in training***, and plotted the results in Figure 5.
>
> Furthermore, we ***removed all intermediate annotations*** and re-conducted the data synthesis and process (see Table 1). Under this setting, the EM score drops by 4.1% compared to using evidence-augmented synthetic data, but ***still surpasses the best baseline by 10.6%***.
>
> ### Reply to Question 3:
>
> While online reinforcement learning can be beneficial in some settings, the development of RAG agents is not inherently dependent on it. For instance, the post-training pipelines of early LLMs such as ***GPT‑3.5 and GPT‑4*** primarily relied on offline SFT and RLHF. ***Despite the absence of online RLVR algorithms (e.g., GRPO) during training, these models already exhibit strong agentic capabilities.***
>
> In the context of this work, the term “***agentic reasoning***” capabilities refer to the “*planning*”, “*state update w.r.t. environmental feedback*” and “*retriever call*” capabilities to be demonstrated at “*inference time*”. Our “open-domain” QA experiments show that ***the policies learned from our offline SFT data can be transferred and executed in an online fashion***.
>
> In contrast, a workflow RAG system utilizes a pre-defined, hard-coded sequence of steps, for example, iteratively retrieve top-k relevant contexts, rerank, summarize and answer, which does not contain the high-level planning and parallel sub-question resolution that our RAG agents have the capacity to perform.
>
> We clarified the distinction between *offline training* and *online execution* in ***Section 5.2 and Appendix E*** of our revised manuscript.
>
> We hope our rebuttal can address your concerns. We are looking forward to receiving your feedback.

---

> > ### Comment · Reviewer_SxGi · 2025-11-27
> > **Official Comment by Reviewer SxGi**
> >
> > Thank you for your efforts and the detailed clarifications. The additional experiments and explanations help address some of my concerns, and I appreciate the revisions you have made.
> >
> > However, several issues remain insufficiently resolved:
> >
> > 1. Novelty and rigor of the abductive reasoning contribution
> > The current implementation still largely resembles LLM-based trajectory reconstruction without a formal notion of abductive optimality or minimality. The revised explanation does not fully clarify how the method differs conceptually from existing structured data-synthesis approaches beyond terminology.
> >
> > 2. Justification of the 0.9 cosine-similarity threshold
> > The rationale remains mostly post-hoc. It would be valuable to include sensitivity analyses across different thresholds, as well as measurements of how threshold variation affects evidence recall/precision and the downstream trajectory structure.
> >
> > 3. Fairness of comparisons due to inconsistent LLM backbones across baselines
> > Many baseline results are taken directly from prior papers, and their LLM backbones differ significantly from those used by EviPath. This limits the fairness and interpretability of the reported performance gaps. It would strengthen the comparisons if the authors could either re-run key baselines under a controlled backbone setting or provide an explicit discussion on how backbone discrepancies may influence the relative strengths of the methods.

---

> ### Author Response · Authors · 2025-12-01
> **Response to the further comments from Reviewer SxGi**
>
> Thank you for your time and effort in providing valuable feedback and suggestions.
> We are happy to offer further clarification regarding your comments for our rebuttal.
>
> **1a. *Regarding the notion of “abductive optimality” or “minimality”***
>
> In our revised manuscript, we no longer claim our abductive trajectory synthesis method is “optimal”. In fact, ***the datasets do not provide ground truth question decomposition or plan*** to solve the problem, ***making the validation of optimality infeasible***.
>
> However, the ground truth evidence associated with each question is the “minimal” information required to answer that question. The ***minimality*** mentioned above in our rebuttal ***refers to the ground truth evidence***, but not our method.
>
> ***1b. The key difference between existing structured data-synthesis approaches and EviPath.***
>
> (i)	Existing structured data-synthesis approaches like LongFaith and CARE can only produce static chain-of-thought rationales. In contrast, ***synthetic trajectories produced by EviPath are “interactive”***, which contain explicit planning steps, retriever calls, or Knowledge Graph Searches.
>
> (ii)	EviPath not only ***guides LLMs on how to use supporting facts***, but also ***teaches how to retrieve key supporting facts from external environments***.  RAG agents trained on our synthetic trajectories can be ***deployed in open-domain scenarios***.
>
> ***2. The rationale of using 0.9 as the evidence selection threshold.***
>
> We have conducted sensitivity analysis for evidence selection threshold $\tau$. Specifically, we select 10000 sub-questions from each of the three datasets and calculate the cosine similarity between their answers and the supporting(golden) or negative facts. ***Figure 6 in Appendix E*** shows that ***most supporting facts have a cosine similarity of around $0.9$ with the sub-question answers***. In the MuSiQue dataset, the similarities between both supporting facts and negative facts and the answers are generally lower, whereas in the 2WikiMQA dataset they are overall higher.
>
> We observe that some sub-questions have low overall similarity, resulting in no evidence meeting the threshold. In such cases, we select the evidence with the highest cosine similarity as the “golden evidence.” Based on this grounding strategy, we compute the ***precision, recall, and F1*** score of evidence selection under different thresholds (see the table below). ***The results in the table below further support the choice of $0.9$ as the threshold.*** *Lowering this threshold reduces the precision of labeling supporting facts for sub-questions.*  For the 2WikiMQA dataset, increasing the threshold beyond $0.9$ could potentially yield higher evidence-labeling precision. However, we would like to adopt a more conservative threshold to balance the performance on different datasets.
>
> |                  |  |   HotpotQA    |       |  |   MuSiQue    |       |  |    2WikiMQA   |       |
> |------------------|:--------:|:-----:|:-----:|:-------:|:-----:|:-----:|:--------:|:-----:|:-----:|
> | Threshold $\tau$ |   Prec.  |  Rec. |   F1  |  Prec.  |  Rec. |   F1  |   Prec.  |  Rec. |   F1  |
> | 0.6              | 0.276    | 0.987 | 0.431 | 0.688   | 0.97  | 0.805 | 0.077    | 0.999 | 0.143 |
> | 0.7              | 0.603    | 0.961 | 0.741 | 0.884   | 0.936 | 0.909 | 0.256    | 0.999 | 0.408 |
> | 0.8              | 0.854    | 0.915 | 0.883 | 0.914   | 0.916 | 0.914 | 0.71     | 0.968 | 0.819 |
> | 0.9              | 0.895    | 0.89  | 0.892 | 0.916   | 0.915 | 0.915 | 0.934    | 0.958 | 0.946 |
> | 0.95             | 0.896    | 0.888 | 0.892 | 0.916   | 0.915 | 0.915 | 0.958    | 0.952 | 0.955 |
>
> ***3. Why it is impractical to unify the backbone models***
>
> Please refer to our reply to ***Reviewer jQ3j***.
>
> We believe that the clarifications above can resolve the mentioned issues. We have also included contents in this response in our revised manuscript ***(see Appendix E, Appendix F - Discussion)***.

---

### Official Review · Reviewer_NVxW · 2025-11-01

**Soundness:** 2
**Presentation:** 3
**Contribution:** 2
**Rating:** 6
**Confidence:** 3

**Summary:**

This paper proposes EviPath, a framework for synthesizing evidence-anchored reasoning trajectories to train RAG agents. Unlike reinforcement learning methods (which suffer from sparse rewards) or prior CoT synthesis (which lacks environmental grounding), EviPath introduces a three-stage abductive reasoning approach: Abductive Subtask Planning (ASP), Faithful Sub-question Answering (ESA), and Conversational Fine-tuning (CFT). Experiments on HotpotQA, MuSiQue, and 2WikiMultihopQA show consistent improvements.

**Strengths:**

1. Reformulates reasoning path synthesis as an abductive reasoning problem, offering a new angle on data synthesis for agentic LLMs.

2. The proposed Planner–Executor decomposition aligns well with the modular RAG agent architecture and avoids costly RL rollouts.

3. Extensive experimental evaluation and good performance across multiple QA datasets, including both text-based and KG-based ones.

**Weaknesses:**

1. "We are the first to formulate the synthesis of reasoning paths for RAG agents as an abductive reasoning problem". I think this is a bit strong and overclaimed. Previous works have made relative exploration (e.g., hotpotQA)

2. Real RAG agents in deployment must handle dynamic, noisy, and incomplete search environments. Evaluating only on static corpora overestimates stability and underestimates retrieval uncertainty.

**Questions:**

See above.

---

> ### Author Response · Authors · 2025-11-19
> **Response to Reviewer NVxW**
>
> Thank you for your acknowledgement of the strengths and contributions of our manuscript.
>
> ### Reply to Weakness 1:
>
> Regarding the claim:
> ```
> We are the first to formulate the synthesis of reasoning paths for RAG agents as an abductive reasoning problem.
> ```
>
> We would like to clarify that “the synthesis of reasoning path” in our claim refers to “***reconstructing complete reasoning trajectories***”. The ***HotpotQA*** paper you mentioned proposes a multi-hop QA dataset of questions, supporting evidence, and answers for evaluation. However, ***it does not provide any ground-truth intermediate reasoning steps, or any decomposed sub-questions, or any environment interaction actions*** that constitute the reasoning path.
>
> Given *the absence of prior work* that formulates the synthesis of reasoning path as an abductive reasoning problem, we believe our claim is appropriate and does not constitute a weakness of our paper.
>
>
> ### Reply to Weakness 2:
>
> We fully agree that RAG agents in deployment must handle dynamic, noisy and incomplete search environments.
>
> Therefore, ***during evaluation, all our experiments*** (except RQ4) ***adopt an “open-domain” setting***, where the agent is required to ***retrieve relevant information from the noisy and incomplete retrieval environment*** (see our claim in Section 5.2).
>
> In other words,  we did ***not*** “evaluate only on static corpora”. Hence, we can conclude that our experiments *neither* overestimate stability *nor* underestimate retrieval uncertainty.
>
> In order to demonstrate the ***robustness*** of our agents in ***handling incomplete retrieval feedback***, we have conducted some case studies. For details, please refer to ***Appendix H*** of our revised manuscript.
>
> Considering that the above concerns have been addressed, we respectfully ask you to consider raising the evaluation scores.
>
> Thank you again for your time. We are looking forward to receiving your feedback.

---

> > ### Comment · Reviewer_NVxW · 2025-11-20
> >
> > Thank you for your response. My previous concerns have been addressed. I have updated my score accordingly.

---

> > > ### Author Response · Authors · 2025-11-20
> > > **Reply**
> > >
> > > Thank you very much for your kind consideration.
> > >
> > > Regards,
> > >
> > > The Authors

---

### Author Response · Authors · 2025-11-17
**Reply to all reviewers**

Dear Reviewers,

Thank you for your acknowledgement of the contribution of our manuscript and the constructive suggestions.

We would like to clarify that, although our training data are synthesized in simulated environments, ***all experiments*** except RQ4 ***are conducted under an “open-domain” setting***.  (See Section 5.2 for our original claim).

Like all baseline methods, ***during evaluation***, RAG agents developed with EviPath data ***also need to interact with “real environment”***, and *handle noisy/incomplete retrieval results*.

Hence, we believe that the excellent performance reported in our tables demonstrates the good quality of our synthetic data, and hence, proves the transferability of RAG agents to be applied in ***“real-world”***, ***“open-domain”*** scenarios.

In addition, we would like to kindly point out that most of the ablation studies you requested in the review *have already been included in our original manuscript*. These include, the removal of golden evidence (Table 1), the removal of training trajectories for the planner/executor (Table 1), and reducing the amount of training samples (Figure 5).

We hope that our clarifications and experimental evidence help address your concerns about our work, and we would be very grateful if you could consider updating your overall assessment accordingly. We will provide detailed, point-by-point responses to the specific weaknesses and questions in the following sections.

Thank you very much for your time. We are looking forward to receiving your feedback.

Regards,

The Authors

---

### Meta-Review · Area_Chair_rhZR · 2026-01-07

**Summary:**

This paper proposes EviPath, a framework for synthesizing evidence-anchored, interactive reasoning trajectories to train retrieval-augmented generation (RAG) agents. The core idea is to reverse-construct multi-step agent trajectories from question–answer pairs and supporting evidence via an abductive reasoning–inspired process, producing planner–executor interactions suitable for supervised fine-tuning. Across HotpotQA, MuSiQue, and 2WikiMultihopQA, models trained with EviPath consistently outperform a broad range of SFT- and RL-based RAG baselines, with extensive ablations exploring data scale, planner/executor roles, and robustness to missing evidence.

Reviewers broadly agree that the paper is carefully executed, empirically strong, and addresses an important limitation of existing RAG agents: the lack of step-wise, process-level supervision. The synthesis pipeline is well motivated, and the experimental results are considered convincing by most reviewers.

At the same time, the discussion centers on three recurring concerns:
(1) the conceptual rigor and novelty of framing the method as “abductive reasoning,”
(2) the realism and scalability of relying on QA pairs and gold evidence during trajectory synthesis, and
(3) the fairness and interpretability of baseline comparisons, especially across different backbone LLMs and training paradigms.

While most reviewers find that the rebuttal substantially clarifies the intended scope of the method and resolves major misunderstandings, one reviewer (jQ3j) remains unconvinced, primarily due to philosophical objections about scalability, robustness under incomplete retrieval, and evaluation practice. The final assessment therefore hinges less on technical soundness, which is largely uncontested, and more on how the venue weighs empirical effectiveness against concerns about generality and realism.

**Reviewer Concerns:**

Concerns substantially addressed by the rebuttal:

Reliance on golden evidence and simulated environments (Reviewers NVxW, SxGi, jQ3j):
The authors convincingly demonstrate through ablations that EviPath does not critically depend on golden evidence. Removing supporting facts still yields double-digit improvements over strong baselines. Additional case studies and statistics on incomplete retrieval further show that models trained with EviPath retain reasonable robustness when retrieval is noisy or partially missing.

Scalability with respect to QA pairs (Reviewers SxGi, jQ3j):
The rebuttal clarifies that EviPath targets small and mid-scale LLMs for which QA supervision is a minimal and common assumption. New experiments using as few as 10k QA pairs show strong gains, alleviating concerns that the method requires prohibitively large or carefully curated datasets.

Fairness of baseline comparisons (Reviewers NVxW, SxGi, jQ3j):
The authors provide detailed justification of training budgets and effective sample counts, explaining why raw question counts are not a fair metric when comparing to RL-based methods. Crucially, new experiments running strong pipeline RAG baselines (HippoRAG, IRCoT) with GPT-4o show that even with substantially stronger backbones, these methods remain clearly behind EviPath-trained 8B agents. This directly addresses the concern that performance gains stem from backbone discrepancies.

Evaluation setting and “static corpus” concerns (Reviewers NVxW, SxGi):
The rebuttal clarifies that all main evaluations (except a specific research question) are conducted in an open-domain setting with real retrieval interaction. This resolves earlier misunderstandings about over-idealized evaluation conditions.

Concerns partially addressed or still remains:

Conceptual rigor of “abductive reasoning” (Reviewer SxGi):
Although the authors appropriately soften claims of optimality and clarify terminology, the method remains largely an LLM-driven trajectory reconstruction process rather than a formally constrained abductive reasoning algorithm. This limitation is acknowledged but not fully resolved.

OOD generalization beyond existing QA distributions (Reviewer jQ3j):
The authors correctly note that “OOD” is not uniquely defined and that their evaluation follows prevailing practice in agentic RAG research. However, the reviewer’s desire for stronger or qualitatively different distribution shifts (e.g., non-QA domains) remains unmet.

Interpretation of robustness metrics (Reviewer jQ3j):
While the authors provide strong contextualization for the reported 29.8% EM under incomplete retrieval, the reviewer remains unconvinced, largely due to an undefined notion of an “overall standard.” This disagreement appears subjective rather than methodological.

**Reviewer Scores:**

Reviewer NVxW.
Initially gave a marginally above-threshold score with reservations about realism and evaluation. After discussion and rebuttal, explicitly stated that their concerns were addressed and updated their score upward. This reviewer would  remain above the acceptance threshold.

Reviewer SxGi.
Initially scored the paper marginally above threshold but expressed deep concerns about rigor, reliance on gold evidence, and fairness. The rebuttal resolved several empirical issues (evidence reliance, data scale, robustness), but some conceptual reservations about abductive reasoning remain. The reviewer would likely maintain a similar marginally positive score, possibly with slightly increased confidence.

Reviewer jQ3j.
Initially rated the paper as a reject and maintained a negative stance after discussion. While many of the reviewer’s technical concerns (fairness of comparisons, robustness, scalability) were directly addressed with new experiments and clarifications, the reviewer continued to raise largely subjective objections (e.g., undefined “overall standard,” dismissal of prevailing evaluation practices). The score might increase slightly but would likely remain below the acceptance threshold.

---

### Decision · Program_Chairs · 2026-01-26

Reject